# Topological digestion drives time-varying rheology of entangled DNA fluids

D. Michieletto [1,2] ✉, P. Neill[3], S. Weir[3], D. Evans [1], N. Crist[3], V. A. Martinez [1] &
R. M. Robertson-Anderson [3] ✉

Understanding and controlling the rheology of polymeric complex fluids that are pushed out-of-equilibrium is a fundamental problem in both industry and biology. For example, to package, repair, and replicate DNA, cells use enzymes to constantly manipulate DNA topology, length, and structure. Inspired by this feat, here we engineer and study DNA-based complex fluids that undergo enzymatically-driven topological and architectural alterations via restriction endonuclease (RE) reactions. We show that these systems display time-dependent rheological properties that depend on the concentrations and properties of the comprising DNA and REs. Through time-resolved microrheology experiments and Brownian Dynamics simulations, we show that conversion of supercoiled to linear DNA topology leads to a monotonic increase in viscosity. On the other hand, the viscosity of entangled linear DNA undergoing fragmentation displays a universal decrease that we rationalise using living polymer theory. Finally, to showcase the tunability of these behaviours, we design a DNA fluid that exhibits a time-dependent increase, followed by a temporally-gated decrease, of its viscosity. Our results present a class of polymeric fluids that leverage naturally occurring enzymes to drive diverse time-varying rheology by performing architectural alterations to the constituents.

DNA is a model polymeric material that naturally occurs in multiple topologies, including supercoiled circular, relaxed circular (ring) and linear conformations. In cells, enzymes convert DNA from one topology to another to perform diverse biological functions[1,2]. One of the most widespread examples of topological alterations to genome architecture is restriction endonuclease (RE) reactions, such as in the CRISPR-Cas9 system[3]. Specifically, type II restriction endonucleases cleave the DNA backbone at specific restriction sites[4] by acting as catalysts to break the sugar-phosphate DNA backbone at specific sites. Thus, REs introduce an irreversible alteration to DNA that pushes the system out-of-equilibrium, forcing it to relax from its initial state to a new thermodynamic equilibrium with distinct physical properties. While REs are abundant in bacteria and engineered to be routinely employed in cloning, the rheological implications of the action of these enzymes are often overlooked despite their ubiquity[4–6].

At the same time, in the materials and engineering communities, polymer topology has long been appreciated for its ability to confer novel rheological properties to polymeric fluids and blends that can be tuned for commercial and industrial use[7–16]. In particular, end-closure of linear polymers (creating open ring, knotted and linked constructs) and breakage and fragmentation of circular polymers (resulting in linear chains) and their roles in the rheology and dynamics of entangled polymer systems is a vibrant and widely investigated topic of research[9,17–28]. While the dynamics of entangled linear polymers are well described by the reptation model developed

[1]School of Physics and Astronomy, University of Edinburgh, Peter Guthrie Road, Edinburgh EH9 3FD, UK. [2]MRC Human Genetics Unit, Institute of Genetics and Cancer, University of Edinburgh, Edinburgh EH4 2XU, UK. [3]Department of Physics and Biophysics, University of San Diego, 5998 Alcala Park, San Diego, CA 92110, USA. ✉e-mail: davide.michieletto@ed.ac.uk; randerson@sandiego.edu

by de Gennes, Doi and Edwards[29], the extension of this model to circular polymers, with no free ends, is not straightforward; and the extent to which ring polymers form entanglements, and the relaxation modes and conformations available to ring polymers remain topics of fervent debate[30–33].

Moreover, in blends of polymers of distinct topologies, such as ring-linear blends, the role of polymer threading and other topological interactions can lead to emergent rheological and dynamical properties such as increased viscosity, suppressed relaxation, and heterogeneous transport modes[19,27,33–36], compared to monodisperse systems of rings or linear chains. For example, solutions of concentrated ring polymers exhibit lower viscosity than their linear counterparts[18,37–39], while blends of ring and linear polymers exhibit higher viscosity than pure linear chains[38,40,41] and display unique behaviours under extensional stress[27].

Far less understood are the rheological properties of supercoiled polymers, for which DNA is an archetypal example[12]. Previous microrheology studies on semidilute blends of ring and supercoiled DNA have shown that blends exhibit entanglement dynamics at concentrations well below that needed for monodisperse systems of ring or linear polymers to exhibit similar viscoelastic properties[41]. At the same time, simulations of dense supercoiled and ring polymers have shown that supercoiled DNA molecules exhibit faster diffusion and more swollen conformations than their ring counterparts[12].

The rich and surprising behaviour of topologically distinct polymeric fluids, along with the principal role that topology plays in dictating the conformational size, structure, and interactions of the comprising polymers, inspired us to exploit DNA topology and DNA-cutting enzymes as a route to functionalise polymeric fluids with time-varying rheological properties. We use solutions of entangled DNA as our prototypical polymeric material as DNA has been extensively employed as a model system to study polymer physics[12,14,36,37,39,41–52]. Further, DNA is particularly well-suited to study the role of topology on the rheology and dynamics of polymeric fluids as it naturally occurs in supercoiled, ring and linear conformations[12,41]. Finally, conversions from one DNA topology to another, via enzymatic reactions, play critical roles in myriad cellular processes such as replication and repair[2].

To achieve this goal, we couple time-resolved microrheology and gel electrophoresis with Brownian Dynamics (BD) simulations and living polymer theory to show that RE-driven topological changes to supercoiled and linear DNA can induce time-dependent rheology in entangled DNA fluids. We measure how the viscosity of these fluids changes under the action of different types and concentrations of REs and use time-resolved analytical gel electrophoresis to correlate the time-varying viscosity to the DNA topology and conformational size.

Surprisingly, our results reveal that cutting DNA by REs yields markedly topology-dependent changes to the fluid viscosity. While cutting long linear DNA yields an expected decrease in viscosity, cutting supercoiled DNA triggers an increase in viscosity and onset of elasticity (Fig. 1). Armed with these results, we designed judicious cocktails of different REs and DNA to engineer fluids that display a transient increase in viscosity, followed by a temporally gated decrease in viscosity. Importantly, we demonstrate that, for all of our systems, the degree to which the viscosity changes and the timescales over which the rheological changes occur can be precisely tuned by varying the concentrations and types of the different REs and DNA constructs used.

Our approach of harnessing topological conversion to drive time-dependent rheological behaviours builds on recent efforts exploring the connection between polymer topology and fluid rheology[12,18,25,28,32,39,41,53]. Notably, our results shed important new light on the impact of supercoiling of ring polymers on the rheology of ring polymer solutions and ring-linear blends. Moreover, our strategy allows for efficient and precise determination of the dependence of rheology on the relative concentrations of topologically distinct polymers comprising blends. For example, in typical experiments that examine how the ratio of rings and linear chains impacts the rheology of ring-linear blends, each blend is 'man-made' by mixing the two solutions comprising the distinct topologies at a specific ratio, such that each data point is from a different sample in a different chamber[53]. This process is not only susceptible to error through concentration variations and mixing inconsistencies, but the number of different topological ratios that can be investigated is limited by available sample and extended data acquisition times. With our approach, a continuum of topological ratios is achieved via in situ RE digestion of a single sample. By tuning the timescale of digestion to be slow (several hours) compared to the measurement time (minutes), we are able to achieve remarkably high resolution in formulation space, measuring the viscosity at, e.g., 24 different blend ratios over the course of 4 h. Extending the measurement window and/or slowing the digestion rate further can allow for even greater precision.

Finally, we emphasise that the fluids we explore in this work are pushed out-of-equilibrium by an irreversible change in their topology, such that they are en-route from an initial equilibrium state to a new one. It is, in fact, this thermodynamic relaxation that we exploit to drive time-dependent alterations to the rheological properties of the DNA fluids. While the final state is known, and, like the initial state, is in thermodynamic equilibrium, the concentrations of the enzymes and DNA, as well as the size and topology of the DNA in the initial state can be precisely tuned to elicit a broad range of rheological transitions that can occur on tunable timescales from minutes to hours.

Our future works will build on this framework by incorporating energy-consuming topological processes and reversibility into the fluids, as well as coupling the fluids to other synthetic and biological systems to pave the way for diverse applications from drug delivery, to filtration and sequestration, to self-repairing infrastructure.

## Results and discussion
### Digestion of entangled circular DNA increases fluid viscosity
We first examine the rheological implications of digesting concentrated solutions of circular 5.9 kbp DNA (pYES2) with the restriction endonuclease, BamHI, that cuts supercoiled (SC) and relaxed circular (R or ring) DNA molecules at a single recognition site to convert them to linear topology (Fig. 1a). We use pYES2 as our substrate as its characterisation and purification have been thoroughly described and validated in previously published works[37,44,54] (also see Supplementary Fig. 4), such that we can be confident in the initial state of the fluids. Likewise BamHI is an extensively used inexpensive RE with validated single-cutting action on pYES2[54]. To quantify how digestion affects the rheology of the fluid, we perform time-resolved microrheology by tracking 1 μm microspheres for ~2 min in 10-min intervals over the course of 4 h (see Methods). Representative bead trajectories in solutions with and without BamHI are dramatically different, with the bead diffusing through a much larger region of space in the absence of the RE compared to in a solution of fully digested DNA (Fig. 1b). To characterise this RE-driven change in mobility, we compute the 1D mean-squared displacement (MSD) from 2D trajectories by averaging the MSDs in the $x$ and $y$ directions separately (Fig. 1c and Supplementary Fig. 1). The magnitude of the MSD decreases monotonically during RE activity, indicating increasing viscosity.

To quantify this behaviour, we compute the zero-shear viscosity $\eta$ via the Stokes–Einstein relation $\eta = k_B T / 3\pi D a$ —with $D = \lim_{t \to \infty} \text{MSD}(t)/2t$ and $a = 1$ μm the diameter of the tracer bead— for each time point $t_a$ during digestion, normalised by the corresponding initial viscosity $\eta(t_a) = \eta_0$. We note that as $t_a$ increases, some of the MSDs transition from exhibiting purely free diffusion, namely MSD $\sim t^\alpha$ with $\alpha = 1$, to displaying modest subdiffusion (i.e., $\alpha \simeq 0.9$) at short lag times (around 0.1 s), suggestive of the onset of high-frequency viscoelasticity.

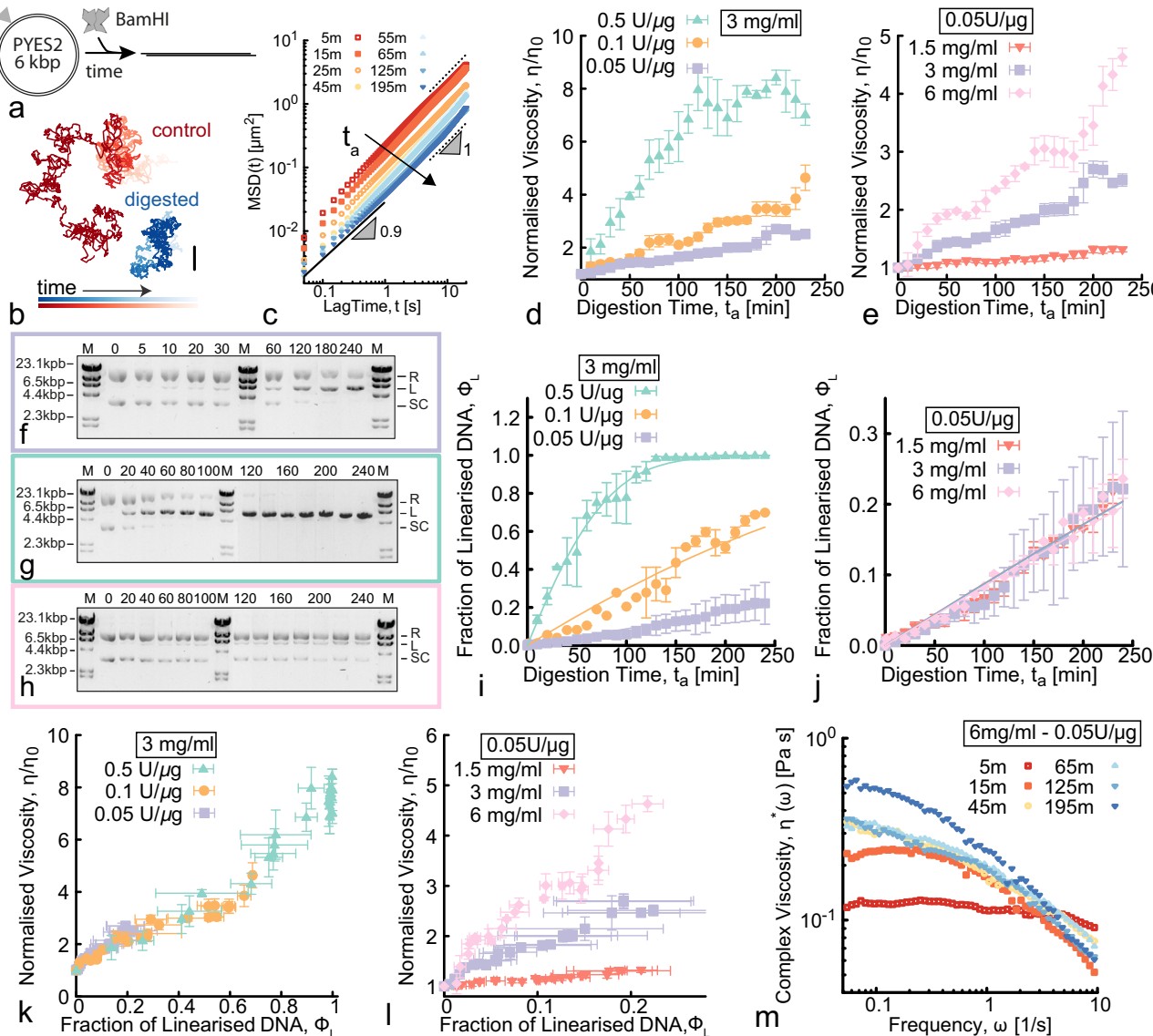

**Fig. 1 | RE-mediated linearisation drives an increase of viscosity of entangled fluids of circular DNA. a** Digestion of 5.9 kbp circular (supercoiled and ring) pYES2 by restriction enzyme BamHI triggers an irreversible architectural change from circular to linear topology. **b** Representative particle trajectories from microrheology in intact (control, red) and digested (blue) solutions. Shading from dark to light indicates increasing tracking time. Scale bar is 1 μm. **c** Mean-squared displacements (MSD) versus lag time $t$ at different digestion times $t_a$ (shown in minutes in the legend) after the addition of BamHI to pYES2 fluids (see Supplementary Fig. 1). The arrow points in the direction of increasing digestion times $t_a$. Black dotted and solid lines represent power-law scaling MSD $\sim t^\alpha$ for free diffusion ($\alpha = 1$) and subdiffusion ($\alpha < 1$). **d, e** Normalised viscosity as a function of digestion time $t_a$ obtained from microrheology for DNA fluids with (**d**) varying RE:DNA stoichiometries at 3 mg/ml, and (**e**) varying DNA concentrations at fixed RE:DNA

stoichiometry of 0.05 U/μg (see Supplementary Fig. 2 for control case with no RE). **f–h** Time-resolved gel electrophoresis taken at different digestion times $t_a$ (listed in minutes above each lane). Border colours correspond to the associated data points. The gels display supercoiled (SC), ring (R) and linear (L) bands of equal DNA length. The marker (M) is the $\lambda$-HindIII ladder (see Supplementary Fig. 3 for the other gel images). **i, j** Fraction of linearised DNA versus $t_a$ for different RE stoichiometries (**i**) and DNA concentrations (**j**) as determined by quantitative analysis of the gels (see Supplementary Fig. 4). The solid lines are fits assuming Michaelis–Menten (MM) kinetics. **k, l** Normalised viscosity as a function of linearised DNA fraction $\phi_L$. The viscosity grows as a function of $\phi_L$ irrespective of RE stoichiometry (**k**), while the growth rate depends on DNA concentrations (**l**). **m** Complex viscosity $\eta^*(\omega)$ versus frequency $\omega$ during digestion of the highest concentration (6 mg/ml) DNA fluid. Error bars represent standard error.

To better characterise potential viscoelastic behaviour, we compute the frequency-dependent complex viscosity $\eta^*(\omega)$ computed from the MSDs using the generalised Stokes–Einstein relation (GSER)[55,56] (detailed in the Methods). While most of the solutions exhibit largely Newtonian behaviour during digestion, manifesting as frequency-independent $\eta^*(\omega)$, the 6 mg/ml solution exhibits high-frequency viscoelasticity for $t_a \geq 15$ min, reflected by the shear-thinning behaviour (i.e., a decrease of $\eta^*(\omega)$ with $\omega$) that increases with increasing $t_a$ (Fig. 1m). In the cases in which we observe high-$\omega$ viscoelasticity, we restrict our analysis to $\omega$ and $\Delta t$ values in which $\eta^*$ is $\omega$-independent and MSDs scale linearly with lag time.

Importantly and intriguingly, we highlight that the observed increase in viscosity during digestion is in marked contrast with the fluidisation of DNA solutions typically observed during digestion[5,6]. Perhaps the first quantitative record of this phenomenon was reported in 1970, when Welcox and Smith used an Ostwald viscometer to measure the change in viscosity of a solution of viral P22 DNA mixed with Haemophilus Influenzae lysate[57]. They measured that the solution became less viscous over time, strongly suggesting the existence of a "restriction" enzyme within the bacterium lysate that was cutting the P22 DNA. This was then identified as HindII, the first restriction enzyme ever discovered and for which Smith was awarded a Nobel

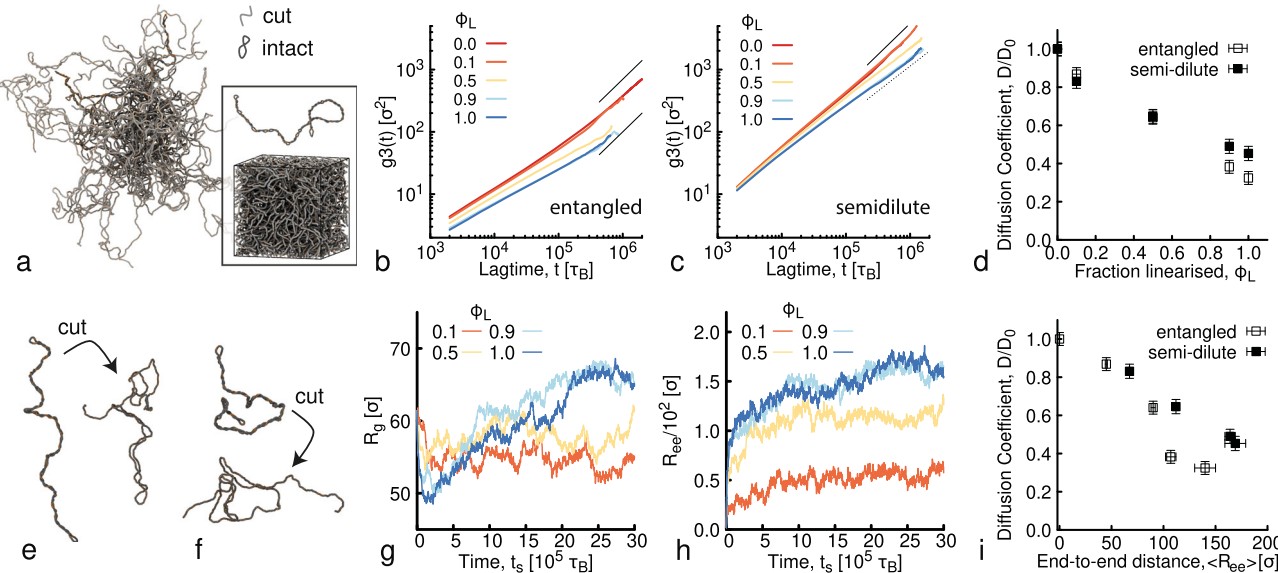

**Fig. 2 | Molecular Dynamics simulations rationalize the observed increase in viscosity during cutting of DNA plasmids. a** Simulation snapshot of entangled 6 kbp ($M = 800$ beads) DNA in which 90% of molecules are linearised ($\phi_L = 0.9$, cut, light grey) and 10% ($\phi_L = 0.1$) remain supercoiled (intact, dark grey). Boxed in is the same system in the simulation periodic box with a single uncut plasmid above it. **b, c** MSD of the centre-of-mass of the chains $g_3(t)$ as a function of lag time $t$ for varying fractions of linearised DNA $\phi_L$ (listed in the legends and serving as a proxy for digestion time $t_a$) in fluids that are (**b**) entangled (volume fraction $\Phi = 4\%$ or $\Phi/\Phi^* \simeq 16$ with $\Phi^* = 0.26\%$[12]) or (**c**) semidilute (volume fraction $\Phi = 0.24\% \simeq \Phi^*$). $g_3(t)$ and $t$ are in simulation units equivalent to $\sigma^2 = 6.25$ nm$^2$ and $\tau_B \simeq 0.03$ µs (see Methods). **d** Diffusion coefficients $D$ determined as $\mathbf{D} = \lim_{t\to\infty} \mathbf{g_3(t)}/6\mathbf{t}$ and normalised by the corresponding value at $t_a = 0$ ($D_0$) for entangled and semidilute

fluids, showing a monotonic slowing down with increasing linearised fraction $\phi_L$. Higher DNA concentration results in a stronger decrease in mobility with increasing $\phi_L$, similar to experiments. **e, f** Snapshot of simulated chains before and after being cut, showing examples of long-lived coiled conformations in entangled (**e**) and semidilute (**f**) conditions. Notice that chains display an initial reduction in conformational size followed by an expansion as the polymer relaxes to steady-state. **g** Radius of gyration $\mathbf{R_g} = \langle\mathbf{R_g^2}\rangle^{1/2} = \langle 1/N\sum_i^N [r_i - r_{CM}]^2\rangle^{1/2}$ and **h** end-to-end distance $\mathbf{R_{ee}} = \langle\mathbf{R_{ee}^2}\rangle^{1/2} = \langle[r_1 - r_N]^2\rangle^{1/2}$ averaged over all (cut and intact) rings in the system. **i** Diffusion coefficients $D/D_0$ plotted against average end-to-end distance $R_{ee}$ measured at large simulation times showing a direct correlation between slower dynamics and larger coil sizes that is stronger for higher DNA concentrations, as seen in experiments.

Prize in Medicine[4]. Since then, DNA digestion has been commonly associated with a decrease in solution viscosity[6]. It is thus quite notable that our solutions of circular DNA cut at only one site present such marked increase in viscosity.

To shed light on this result, we examine how RE concentration tunes the magnitude and rate of the increase in viscosity. In Fig. 1d, we show that the rate of viscosity increase scales with RE concentration, indicating that this behaviour is directly linked to the conversion of circular DNA into linear topology. For the highest RE concentration, a time-independent plateau is reached in < 2 h, suggestive of complete digestion and arrival at a new equilibrium (Fig. 1d). Conversely, for lower RE concentrations, $\eta/\eta_0$ continues to steadily increase over the time course of the experiment, following a roughly linear trend, indicating that the fluid is out-of-equilibrium, relaxing to its new equilibrium, for the duration of the experiment.

**Higher DNA concentrations elicit more pronounced increase in viscosity**

Less intuitive than the effect of RE concentration, is how DNA concentration may impact the time-varying rheology at a fixed RE:DNA stoichiometry. Indeed, while increased viscosity can slow the rate of enzymatic processes, macromolecular crowding can increase some enzymatic reactions[58–60]. Figure 1e shows that DNA concentration has a pronounced impact on the degree to which the viscosity increases: $\eta/\eta_0$ for the 6 mg/ml solution increases ~5-fold during digestion, whereas there is a nearly undetectable increase for the 1.5 mg/ml solution.

To quantitatively link the time-dependent increase in viscosity to the topological changes of the DNA molecules we perform time-

resolved analytical gel electrophoresis, as described in Methods and Supplementary Figs. 3–6. The gels shown in Fig. 1f–h, which can be interpreted based on the reference bands and characterisation shown in Supplementary Fig. 4, demonstrate that supercoiled (SC, bottom band) and relaxed circular (R, top band) populations are converted to linear (L, mid band) topology over the course of ~4 h. By using quantitative band intensity analysis (Supplementary Figs. 3 and 4), we find that the rate of linearisation, $d\phi_L/dt$, increases with increasing RE concentration (Fig. 1i), as expected. On the contrary, DNA concentration $c$ has undetectable impact on the digestion rate (Fig. 1j). In both cases, the fraction of linearised molecules $\phi_L$ pleasingly follow Michaelis–Menten kinetics, with roughly linear increase at short times followed by a slowing down as substrates (i.e., uncut plasmids) get depleted (see solid curves in Fig. 1i, j). The Michaelis constant that we obtain with this analysis is in agreement with ~0.5 µM reported in the literature for the RE used in our experiments, BamHI[61].

To directly relate the change in topology with the increase in viscosity, we plot $\eta/\eta_0$ versus $\phi_L$ (Fig. 1k, l). These plots clearly show that the viscosity of our DNA samples not only depend on the degree of linearisation (i.e., $\phi_L$), but also on the entanglement density, which depends on both $c$ and $\phi_L$.

Importantly, the increase in viscosity shown in Fig. 1k, l is generally monotonic over the course of the digestion time, counter to the non-monotonic dependence of viscosity reported in ring-linear polymer blends[38,40,41]. However, the trend is 'noisy', with dips and valleys for all but the lowest DNA concentration, as well as larger standard deviations in the data at each time point. Increased dynamical heterogeneity has been previously predicted and observed in entangled rings and ring-linear blends[45] and attributed, in part, to

ring-ring and ring-linear threading events[36]. Other features of our data that support the contribution of threading to the rheology are the steep upticks in the viscosity for the (6 mg/ml, 0.05 U/μg), (3 mg/ml, 0.05 U/μg) and (3 mg/ml, 0.10 U/μg) cases, which occur at times that correspond to ~15% and ~70% linear chains. The low linear fraction is comparable to the fraction of linear chains needed in ring polymers to markedly reduce ring diffusion via threading[18,19,43,62,63], while the high fraction is similar to that previously reported for a large increase in the complex viscosity[39–41].

Further, our gel electrophoresis analysis (Fig. 1f–h and Supplementary Fig. 3) shows that the supercoiled DNA is digested at a faster rate than the ring DNA, such that we expect ring-linear threading to contribute more to our observations at later times in the digestion. Indeed, the features described above are most apparent at these later times. Finally, we note that while threading of open rings has been well-established, the literature regarding the extent to which supercoiled polymers can become threaded is scarce and lacks consensus[41,64].

## Molecular dynamics simulations link the rise in viscosity to increased conformational size of DNA

To better understand the increase in fluid viscosity seen above we simulate dense solutions of twistable bead-spring polymer chains[12,65], representing 6 kbp plasmids, under the effect of cutting agents (see Fig. 2a). We simulate chains 800 beads long, meaning that each bead represents $\sigma = 7.5$ bp $= 2.5$ nm. The system is run with an implicit solvent (Langevin dynamics) at temperature $T = 1\epsilon/k_B$ (see Methods for more details). We consider entangled and semidilute systems at volume fractions $\Phi = 4\%$ and $\Phi = 0.24\%$, respectively (the overlap volume fraction is $\Phi^* \simeq 0.26\%$) and supercoiling degree $\sigma = 0.06$ (see Methods and Supplementary Note 1 for more details). We start with entirely supercoiled polymers (no relaxed rings) and simulate RE digestion by linearising a fraction $\phi_L = 0$, 0.1, 0.5, 0.9, and 1 of the chains to mimic different time points during digestion.

The linearisation is done by removing one bead from each cut ring together with its patches and all the bonds, angles and dihedrals that it is part of. Simultaneously, we zero all the torsional constraints along the cut chain, but leave the bending rigidity unaltered. This procedure is motivated by recent simulations showing that the twist relaxation of DNA is orders of magnitude faster than the relaxation of the writhe[66]. As such, shortly after being cleaved by a restriction enzyme, DNA is likely torsionally relaxed, yet still displays substantial unresolved writhe, as in our simulations (Fig. 2). After cutting, we let the system equilibrate (Fig. 2a) and measure the centre-of-mass MSD of the polymers, $g_3(t) \equiv \langle [\mathbf{r}_{CM}(t_0 + t) - \mathbf{r}_{CM}(t_0)]^2 \rangle$.

As shown in Fig. 2b, we find that, in agreement with our microrheology data (Fig. 1k, l), the larger the fraction of linearised DNA, the monotonically slower the dynamics, which we quantify by computing the diffusion coefficient of the centre-of-mass of the chains at large times $D(\phi_L)$, normalised by its value when there are no cut chains $D_0 = D(\phi_L = 0)$. As shown in Fig. 2d, the reduction in $D/D_0$ is stronger for the higher DNA concentration, as seen in experiments (Fig. 1e, l), with the effect being most pronounced at higher $\phi_L$ (corresponding to longer digestion times).

To shed light on the topological effects that give rise to this slowing down, we measure the radius of gyration $R_g$, averaged over all simulated chains, as a function of simulation time $t_s$ (Fig. 2g). As shown, immediately after cutting, there is a drop in $\langle R_g \rangle$ as the supercoiled linear-like[12] configurations begin to unravel and segments adopt more entropically favourable configurations (Fig. 2e). As the chains continue to unravel they once again swell as they assume random coil configurations (Fig. 2f). This non-monotonic conformational uncoiling is most evident at $\phi_L = 0.9$ and $\phi_L = 1$, where the polymer dynamics are slow and complete equilibration is not reached until the last ~30% of the simulation time, signified by the time-independent plateau in $\langle R_g \rangle$ (Fig. 2g). We also compute a

complementary metric of the conformations assumed by the chains, the average root-mean-square end-to-end distance $\langle R_{ee} \rangle$, which shows similar trends as $\langle R_g \rangle$ (Fig. 2h).

To directly correlate dynamics with polymer conformations in our simulations we plot $D/D_0$ as a function of $\langle R_{ee} \rangle$ (Fig. 2i), which clearly shows that the increase in coil size dictates the slowing down of the dynamics. Importantly, the degree to which diffusion is slowed with increasing $\langle R_{ee} \rangle$ is stronger for higher DNA concentrations, in line with experiments (Fig. 1i).

To directly compare our simulations and experiments, we evaluate the experimentally measured diffusion coefficients for the 1 μm beads as a function of $\phi_L$ and compare the corresponding $D/D_0$ values with the results from simulations shown in Fig. 2. As shown in Fig. 3a, both experimental and simulated diffusion coefficients drop by ~2–5-fold as $\phi_L$ increases to 1, and this decrease is larger for higher DNA concentrations. Simulations do show a slightly weaker dependence of $D/D_0$ on $\phi_L$ and concentration, as compared to our experimental data, which we conjecture arises from the lack of open rings in the simulations. Recall that our experimental solutions start with both supercoiled and open rings, with the rings appearing to be cut at a slower rate (Fig. 1f–h). As such, we attribute the less pronounced slowing down in the simulations compared to experiments to fewer threading events, which have been shown to markedly slow dynamics[12,63].

As our simulations indicate that the changing conformational size of the polymers, as well as the degree of entanglement (i.e., $c$), are driving factors in the increasing viscosity we measure experimentally, we next compute $\langle R_g \rangle$ as a function of time for the data shown in Fig. 1. We use previously reported $R_g$ values and relations $R_{gl}/R_{gR} \simeq 1.6$ and $R_{g,L}/R_{g,SC} \simeq 2.1$[37,41], along with $\phi_L$ values quantified from gel electrophoresis analysis (Fig. 1i, j), to compute $\langle R_g \rangle$, averaged over all topologies, as a function of digestion time $t_a$ and linearised fraction $\phi_L$. As shown in Supplementary Fig. 5, $\langle R_g \rangle$ increases with $\phi_L$, similar to simulations. The impact of increasing $\langle R_g \rangle$ on the dynamics is captured in Fig. 3b, in which we plot $D/D_0$ as a function of $\langle R_g \rangle$. As shown, both experimental and simulated $D/D_0$ values decrease with increasing $\langle R_g \rangle$ with remarkable agreement. Notably, both sets of data show that $D$ decreases more rapidly with $\langle R_g \rangle$ for higher DNA concentrations.

To rationalize the concentration and size dependence we discuss above, we recall that in semidilute and concentrated solutions, changing $R_g$ directly alters the degree of overlap via the relation $c^* = 3M/(4\pi N_A R_g^3)$, where $c^*$ is the polymer overlap concentration. To determine the role of topology-driven changes in polymer overlap, owing to the changing size of the polymer coils, we plot the normalised viscosity $\eta/\eta_0$ against $c/c^*$. Note that $c/c^*$ will decrease if $c$ decreases or if $R_g$ increases (as it does during digestion). To compute $c^*$ during digestion, in which we have a mix of different topologies —each with a different $R_g$— we start with the conventional expression $c^*$, derived by equating the solution volume ($m/c^*$, where $m$ is total mass) to the total volume the molecules comprise, i.e., the total number of molecules ($N = mN_A/M$) multiplied by the volume per molecule ($V_m = 4\pi R_g^3/3$). We then consider that each component contributes separately to the total volume the molecules fill in solution: $N_S V_{m,SC} + N_R V_{m,R} + N_L V_{m,L} = (4\pi/3)N(\phi_R R_{g,R}^3 + \phi_S R_{g,SC}^3 + \phi_L R_{g,L}^3)$ where $\phi_R + \phi_{SC} + \phi_L = 1$. The resulting expression is then: $c^* = (3/4\pi)M/N_A(\phi_R R_{g,R}^3 + \phi_S R_{g,SC}^3 + \phi_L R_{g,L}^3)$.

As shown in Fig. 3c, our normalised viscosity data $\eta/\eta_0$ all collapse onto a master curve when plotted against $c/c^*$, with a functional form that can be described by $\eta/\eta_0 \sim (c/c^*)^\gamma$, where $\gamma = 0.5$ and $\gamma = 1.75$ at low and high $c/c^*$, respectively. The crossover in scaling takes place at $c \simeq 6c^*$, which we previously showed to be the critical entanglement concentration for DNA solutions[37]. The exponents are in agreement with those reported in single-molecule tracking studies measuring the diffusion of ring and linear DNA in semidilute ($c \simeq c^*$) and entangled ($c \gtrsim 6c^*$) DNA solutions, respectively[44]. Specifically, these studies

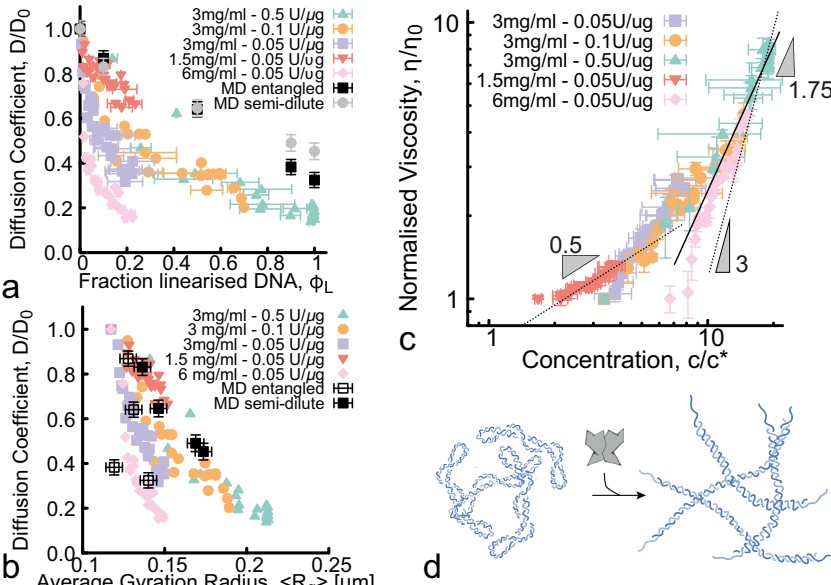

**Fig. 3 | Mapping changes of topology and conformational size onto fluid viscosity. a** Normalised diffusion coefficients $D/D_0$ versus fraction of linearised DNA $\phi_L$, as measured via microrheology (coloured data points) and MD simulations (black data points), show slowing of DNA mobility with increasing $\phi_L$ that is stronger for higher DNA concentrations. The discrepancy between experiments and simulations may be due to the fact that experiments have a population of relaxed ring DNA not present in simulations, which may be more prone to threadings. **b** Data shown in **a** plotted against the average radius of gyration $\langle R_g \rangle$ measured in experiments (colours) and simulations (black), demonstrating that

the decrease in mobility correlates with increased conformational size and that the slowing is more pronounced at higher DNA concentrations. **c** Experimentally measured normalised viscosity $\eta/\eta_0$ plotted against DNA concentration $c$, normalised by the overlap concentration $c^*$. Dotted and solid lines show scaling relations $\eta/\eta_0 \sim (c/c^*)^\gamma$ with $\gamma$ values corresponding to Rouse-like diffusion of semidilute polymers ($\gamma = 0.5$)[67], reptation of entangled polymers ($\gamma = 1.75$)[68], and constraint release of threaded circular polymers ($\gamma = 3$)[14,81]. **d** Cartoon depiction of increased viscosity caused by weakly entangled supercoiled polymers being cut by REs to form heavily entangled linear chains due to increasing $\langle R_g \rangle$ and lowering $c^*$.

showed that $D \sim c^{-1.75}$ for 11, 25 and 45 kbp ring and linear DNA at concentrations above $\sim 6c^*$, in line with reptation model predictions[67,68]. Below $\sim 6c^*$, diffusion coefficients followed $D \sim c^{-0.5}$ scaling, in accord with Rouse model predictions[67,68]. We note that for the highest concentrations ($c/c^* > 10$), the scaling of $\eta/\eta_0$ is steeper than $\gamma = 1.75$ and more closely follows $\gamma = 3$. This result aligns with previously reported scaling for ring-linear DNA blends in which constraint release from threading of rings by linear chains dominated the diffusivity[14,38,44]. Taken together, our experiments and simulations indicate that the observed RE-driven increase in viscosity arises from the conversion of circular DNA to linear topology, which in turn increases the average conformational size and entanglement density (Fig. 3d).

**Digestion of entangled linear DNA drives a universal decrease in fluid viscosity**

Having demonstrated that REs can enable a tunable increase in viscosity of DNA fluids over time, we now turn to designing and understanding DNA fluids that decrease their viscosity in time. We use entangled linear $\lambda$-DNA (48.5 kbp), which is $\sim 8x$ larger than the 5.9 kbp circular DNA we study above. We start with a high degree of entanglements ($c = 1$ mg/ml, $c/c^* \simeq 20$, $c/c_e \simeq 3$)[69,70] and examine the rheological implications of digestion by various REs that cut the DNA into 5, 6 or 141 linear fragments that have either ends that are blunt or have single-strand 'sticky' overhangs (Fig. 4a and Supplementary Table 1). Importantly, this commercially available DNA construct has been used for decades as a model polymer system and has been exhaustively characterised in the literature[46,70–72], such that we can be confident of the initial properties of the fluid. Moreover, our analytical gel electrophoresis experiments Fig. 4j show that both the initial DNA length, as well as the number and lengths of the digested fragments are as expected, further indicating the structural purity of our $\lambda$-DNA fluids.

Figure 4b shows that representative trajectories without REs are dramatically more confined than those undergoing digestion by the

RE, HindIII, that cuts $\lambda$-DNA at five distinct sites. Correspondingly, the MSDs become consistently faster during digestion, indicating that fragmentation is reducing the entanglement density (Fig. 4c). Analogous to what we see in Fig. 1c, some MSDs display a subdiffusive regime at short lag times $t$, reflecting the viscoelasticity of entangled $\lambda$-DNA[69,71]. MSDs crossover to free diffusion after $t \simeq 2$ s, comparable to the longest relaxation time (i.e., the reptation or disengagement time $T_d$) of our $\lambda$-DNA solutions[69]. On the contrary, fully digested fluids display free diffusion across all lag times, suggesting minimal viscoelasticity (Fig. 4c). Similar to Fig. 1m, we characterise the viscoelasticity as a function of digestion time by computing the complex viscosity $\eta^*(\omega)$ from the MSDs using GSER. Figure 4d indeed shows high-$\omega$ viscoelasticity (evidenced by shear-thinning) that becomes progressively weaker as $t_a$ increases.

As in Fig. 1d, e, we use the Stokes–Einstein relation to extract the normalised zero-shear viscosity $\eta/\eta_0$ from the MSDs at large lag times (low $\omega$), in which viscoelastic effects are negligible (Fig. 4e, f). This analysis reveals that, indeed, REs with $\lambda$-DNA recognition sites specifically decrease the fluid viscosity (see Supplementary Fig. 2 for null control).

We next examine how varying the RE concentration impacts the time-dependent reduction in viscosity $\eta(t_a)/\eta_0$ over the course of the digestion time $t_a$ (Fig. 4e). We use BamHI, which cuts $\lambda$-DNA into 5 smaller fragments with short overhangs, at concentrations of 0.27 to 5.5 U/$\mu$g. As expected, the viscosity decreases more rapidly at higher RE concentration (Fig. 4e). At later times, the faster reactions appear to slow down, consistent with Michaelis–Menten kinetics when the substrate is depleted.

Given the known role that polymer length plays in the rheological properties of polymer fluids, we anticipate that the number, lengths, and type of DNA fragments generated by a given RE will impact the time-dependent reduction in viscosity. Surprisingly, however, we find that the viscosity reduction appears to be broadly RE-independent. The $\eta/\eta_0$ curves obtained by digesting $\lambda$-DNA with five different REs,

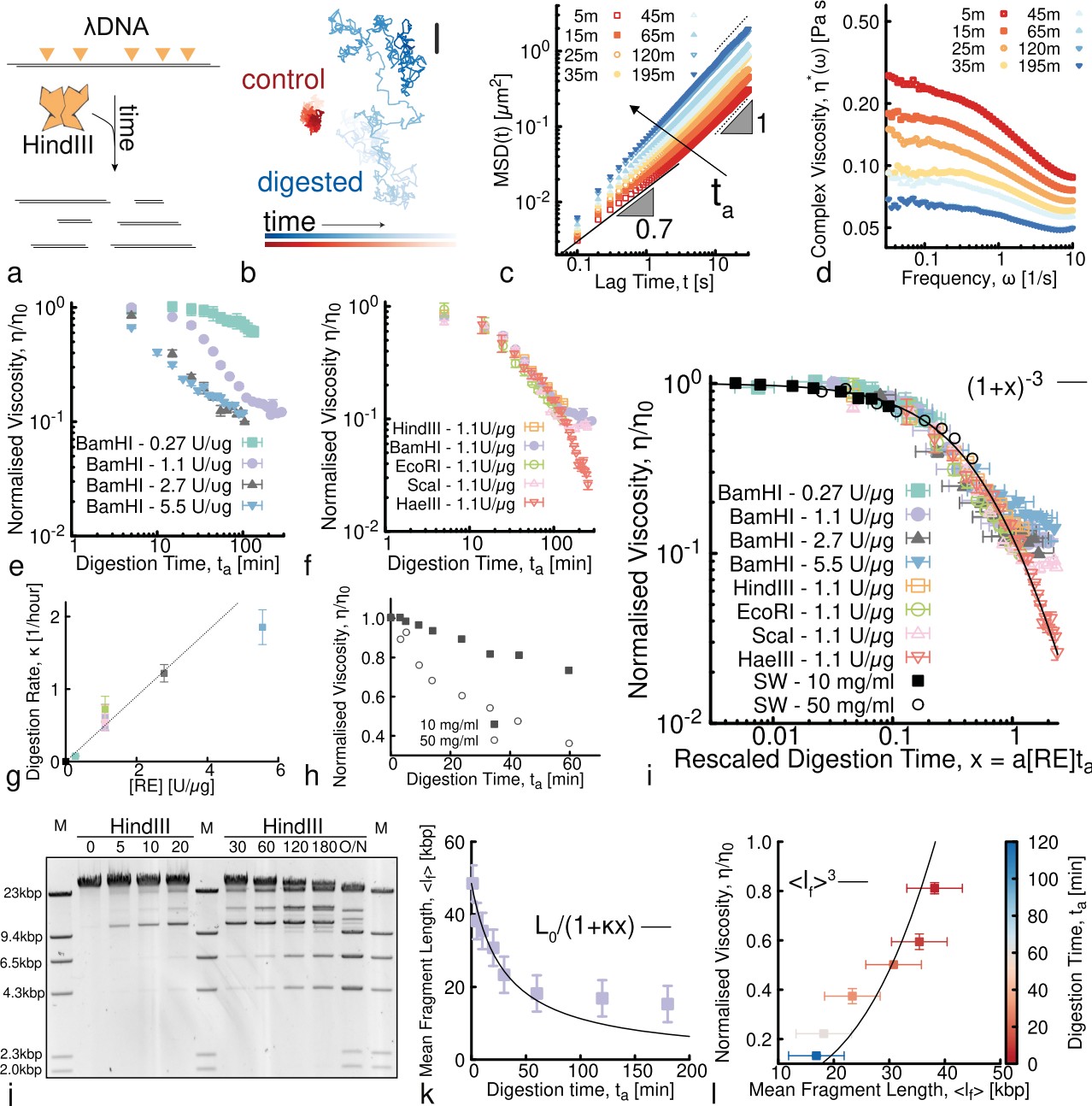

**Fig. 4 | Solutions of entangled linear λ-DNA undergoing digestion display a universal decrease in viscosity. a** Digestion of linear λ-DNA (48.5 kbp) into smaller fragments via a multi-cutter RE HindIII. **b** Representative trajectories from microrheology in inactive (control, red) and digested (blue) solutions. Shading from dark to light indicates increasing tracking time. Scale bar is 1 μm. **c** Mean-squared displacements (MSD) versus lag time $t$ at different digestion times $t_a$ (shown in minutes in the legend) after the addition of 1.1 U/μg of HindIII. Black dotted and solid lines represent power-law scaling MSD ~ $t^a$, with corresponding scaling exponents listed. **d** The complex viscosity $\eta^*(\omega)$ at different digestion times displays a shear-thinning behaviour that weakens at long digestion times. **e, f** Viscosity during digestion time as a function of (**e**) different RE (BamHI) concentrations (listed as RE:DNA stoichiometry in the legend) and (**f**) different RE (fixed stoichiometry 1.1 U/μg) (See Supplementary Table 1 for more information on REs). **g** By fitting the data shown in **e, f** with $(1 + \kappa t)^{-3}$ (see Eq. (2)), we find the

digestion rate to be a linear function of the RE concentration, i.e., $\kappa = a[\text{RE}]$. **h** Data from the Smith and Welcox paper where they discover HindII in 1970[57]. The legend lists the concentration of *Heamophilus Influenzae* lysate. **i** By rescaling digestion time as $t_a \rightarrow a[\text{RE}]t_a$ with $a$ an RE-independent constant, all data collapse onto a master curve predicted by Eq. (2). **j** Time-resolved agarose gel electrophoresis showing digestion of entangled λ-DNA by HindIII. Digestion time $t_a$ (in mins) is listed above each lane and M denotes lanes with the λ-HindIII marker. **k** We quantify the bands intensity to obtain the average fragment length $\langle l_f \rangle$ as a function of digestion time $t_a$. Solid line is a fit of the data to the predicted scaling $L_0/(1 + \kappa t)$. **l** We correlate fluid rheology to DNA architecture by plotting $\eta/\eta_0$ measured for samples digested with HindIII at 1.1 U/μg with $\langle l_f \rangle$ extracted from the gel. The solid line is the theoretical predicted trend $\eta/\eta_0 = \langle l_f \rangle^3$. Error bars represent standard error.

which vary in the number of cleaving sites (5, 6 or 141), fragment lengths, and end type (blunt or overhangs), all collapse to a single master curve (Fig. 4f). Slight enzyme dependence does manifest at large digestion times, $t_a \gtrsim 100\,s$, where most of the curves transition

to very weak time-dependence. The notable exception is HaeIII, which has 141 restriction sites (compared to 5 or 6 for the other REs), suggesting that the nearly time-independent plateaus indicate that the number of available cleaving sites is approaching zero.

## A generalisation of living polymer theory explains the RE-driven decrease in viscosity

To better understand the trends we observe in Fig. 4e, f, we employ a generalisation of equilibrium living polymer theory[73–79]. First, we note that there is a substantial separation between the longest polymer relaxation timescale $T_d$ and the RE cleaving (digestion) timescales $\tau_c$. More specifically, the relaxation of $\lambda$-DNA at 1 mg/ml is ~2 s[69] whereas the digestion timescale considered in this work is of the order of tens of minutes. The non-dimensional parameter $\chi \equiv T_d/\tau_c \lesssim 10^{-3}$, that captures the average number of cleaving reactions occurring within one polymer relaxation timescale, indicates that our fluids are in structural quasi-equilibrium during the microrheology measurements we perform ($\chi \ll 1$).

Within this quasi-steady-state regime, we compute the stress relaxation of a chain as the survival probability of its tube segments, $\mu(t)$. For monodisperse and entangled polymer systems this probability can be approximated as $e^{-t/T_d}$[68], where $T_d(L_0) = L_0^2/(D_c\pi^2)$, $D_c \sim D_0/L_0$ is the curvilinear diffusion coefficient, and $D_0$ is a microscopic diffusion constant[68]. The stress relaxation can then be found as $G(t) = G_0\mu(t)$, with $G_0$ the instantaneous shear modulus, and the zero-shear viscosity is $\eta_0 = \int_0^\infty G(t)dt \simeq G_0 T_d$. For entangled linear DNA undergoing digestion, irreversible cleavage must be taken into account in the chain relaxation process. The action of REs thus drives the system into a polydisperse state with mean length $\langle l_f \rangle$ that depends on the digestion time $t_a$. A mean field calculation yields

$$\langle l_f(t_a) \rangle = \frac{L_0}{(n_f(t_a)+1)} = \frac{L_0}{(\chi t_a/T_d+1)}, \qquad (1)$$

where $n_f(t_a) + 1 = \chi t_a/T_d + 1 = t_a/\tau_c + 1$ is the average number of DNA fragments at digestion time $t_a$. The typical relaxation timescale at time $t_a$ is that necessary for the relaxation of the average fragment length with instantaneous curvilinear diffusion $D_f(t_a) = D_0/\langle l(t_a) \rangle$, i.e., $\tau_r = \langle l_f(t_a) \rangle^2/D_f(t_a) = (L_0^3/D_0)(\chi t_a/T_d + 1)^{-3}$. For small digestion times (compared with the typical cleavage time $\tau_c = T_d/\chi$) the system behaves as if made by unbreakable chains with relaxation time $T_d \sim L_0^3/D_0$. In the opposite limit, one finds that $\tau_r \sim 1/(D_0(t_a/\tau_c)^3)$ up to times in which the reptation model is no longer valid or the number of fragments plateaus because there are fewer uncleaved sites. Accordingly, the zero-shear viscosity is directly proportional to this relaxation timescale, such that:

$$\eta(t_a) \simeq G_0\tau_r = \frac{\eta_0}{(\chi t_a/T_d + 1)^3}, \qquad (2)$$

with $\eta_0$ the zero-shear viscosity before digestion begins.

Eq. (2) predicts that (i) the viscosity should decrease over time irrespective of RE (as seen in our experiments, Fig. 4f), (ii) the key digestion rate $\kappa = \chi/T_d = 1/\tau_c$ should be proportional to RE concentration (as seen in Fig. 4g), and (iii) the normalised viscosity should scale as $(1 + \kappa t_a)^{-3}$. By fitting each of our experimental normalised viscosity curves in Fig. 4e, f with the function $\eta/\eta_0 = (1 + \kappa t_a)^{-3}$, we obtain a direct measure of the corresponding digestion rate $\kappa$, which appears to be linear with RE concentration (Fig. 4g) in agreement with Michaelis–Menten theory. Pleasingly, upon rescaling time by the corresponding RE-independent $\kappa$ value (i.e., $t_a \rightarrow \kappa t_a = x$), all of our experimental curves collapse onto a single master curve described by $\eta/\eta_0 = (1 + x)^{-3}$ (Fig. 4i). To further demonstrate the generality of our theory, we include the data presented in Smith and Welcox's seminal 1970 paper[57] (Fig. 4h) and show that these data also collapse onto the same master curve upon rescaling time as $t_a \rightarrow bc_l t_a$ with $b$ a constant and $c_l$ the lysate concentration.

Finally, to directly couple the observed decrease in viscosity to the RE-mediated architectural alterations, we perform time-resolved analytical gel electrophoresis, as shown in Fig. 4j. We analyse the intensity of each band, which corresponds to a fragment of a specific length, to quantify the concentration of each digested fragment (as described in Supplementary Fig. 4) from which we compute the average fragment length $\langle l_f \rangle$ as a function of digestion time $t_a$ (see Methods and ref. 80). As shown in Fig. 4k, the average fragment length exhibits an initial rapid drop in $\langle l_f \rangle$ followed by a slower decay, compatible with the predicted relation described above, $\langle l(t_a) \rangle = L_0/(1 + \kappa t_a)$ where $1 + \kappa t_a$ is the average number of fragments at time $t_a$. A crossover to slower decay rates occurs at ~45 min, which is nearly identical to the time at which we see the switch to weaker time-dependence in experiments (Fig. 4d). We correlate the measured viscosity with the structural data by plotting $\eta/\eta_0$ against $\langle l_f \rangle$ at different digestion times $t_a$ (Fig. 4l). The curve displays a functional form compatible with $\langle l_f \rangle^3$ expected for reptation. Deviation from reptation behaviour for small $\langle l_f \rangle$ is likely due to the fact that when $\lambda$-DNA is digested to short fragments, the reptation model is no longer valid[69,70].

## Engineering viscosity gating and non-monotonicity into the time-dependent rheology of DNA fluids

In the preceding sections, we demonstrate that the action of REs can either increase and decrease the viscosity of entangled DNA fluids over time, depending on the initial DNA topology and type of RE. We show that the rate and degree to which the viscosity changes over time can be tuned by varying the DNA concentration and topology as well as the RE concentration and type.

We leverage these findings to design systems that exhibit non-monotonic time-varying rheology. Specifically, we engineer DNA fluids to display an initial rise in viscosity followed by a decrease, with the rate and magnitude of the changes in viscosity, as well as the time that peak viscosity is reached (i.e., the 'gating time'), tuned by varying the relative concentrations of the REs.

To achieve this design goal we prepare a fluid comprising a mixture of supercoiled and relaxed circular DNA (IE241 plasmids) with length and concentration comparable to our $\lambda$-DNA fluids (44 kbp, 1.4 mg/ml), which we digest with two REs: XhoI, which cuts the DNA twice, and EcoRI, which produces 10 linear fragments (Fig. 5a). The rationale behind these choices is to incorporate properties of both DNA fluids and RE types used in the preceding sections. Namely, starting from a population of supercoiled plasmids that get linearised should exhibit increasing viscosity as in Fig. 1, while adding an RE with many target sites on the chosen plasmid should yield a fluid that exhibits decreasing viscosity, as in Fig. 4.

As expected, cleavage by XhoI, in the absence of EcoRI, increases the fluid viscosity by an order of magnitude over the course of ~30 min (see Fig. 5b, cyan squares). We understand this rheological behaviour to be a result of linearisation of circular DNA, which can also be seen in Supplementary Fig. 6, and is in line with our findings in Fig. 1. We next incorporate a low concentration of EcoRI to trigger a decrease in viscosity that should follow the faster XhoI-mediated increase. As seen in Fig. 5b, c, by varying the concentration of EcoRI from 400x to 2x lower than XhoI, we are able to tune the gating time, rate, and magnitude of the viscosity reduction.

As we increase [EcoRI] further (keeping [XhoI] fixed), we observe a decrease in both the gating time $T_{\text{gate}}$, which we define as the digestion time $t_a$ at which the maximum viscosity is reached, as well as the relative increase in viscosity, $\eta_{\text{max}}/\eta_0$ (Fig. 5b, c). Conversely, the relative decrease in viscosity, which we quantify as $\eta_0/\eta_{\text{min}}$, is more dramatic with increasing [EcoRI]. Indeed the viscosity drops by over an order of magnitude at the highest [EcoRI] (Fig. 5b). This comparatively strong dependence indicates that the viscosity reduction at long times is largely determined by the process of fragmentation, while the height of the viscosity peak at shorter times can be controlled by the concentration and length of the DNA plasmids chosen to make up the fluid.

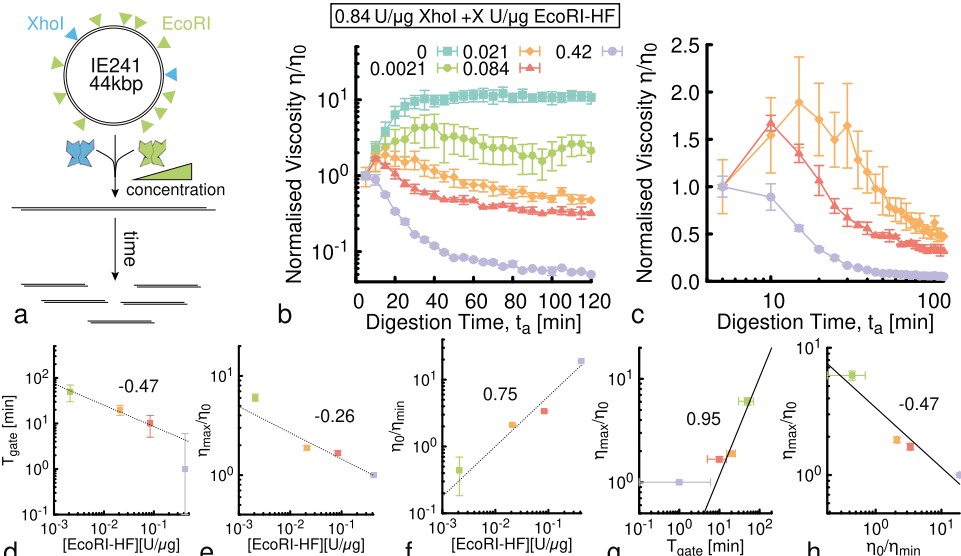

**Fig. 5 | Time-dependent and non-monotonic rheology of entangled DNA fluids.**
**a** Our fluid comprises 44 kbp circular DNA (IE241) and two different REs that cut the DNA twice (XhoI) or 10 times (EcoRI). The green ramp indicates that we vary the concentration of EcoRI while keeping the DNA and XhoI concentrations fixed. **b** Normalised viscosity versus digestion time for solutions of 1.4 mg/ml circular DNA undergoing architectural conversion by a fixed concentration of XhoI (0.84 U/μg) and five different EcoRI concentrations (listed in the legend). Fluids exhibit initial rise in viscosity, due to XhoI-driven linearisation of the circular DNA, followed by a decrease, due to fragmentation by EcoRI. The maximum increase in viscosity, $\eta_{max}/\eta_0$, and time at which the viscosity starts to decrease, i.e., the gating time $T_{gate}$, is dependent on the stoichiometry of the REs. **c** Data shown in (**b**) for the three lowest EcoRI concentrations, zoomed-in and plotted on log-$x$ scale to more clearly show [EcoRI]-dependent behaviour and the gating time. **d** The gating time, which we define as $T_{gate} = \mathrm{argmax}[\eta/\eta_0]$, **e** the maximum increase in viscosity, defined as $\eta_{max}/\eta_0$, and **f** the maximum decrease in viscosity, i.e., $\eta_0/\eta_{min}$, appear to scale with [EcoRI] as power laws, with approximate exponents −0.47, −0.26 and 0.75, respectively. **g** The maximum viscosity scales nearly linearly with gating time, i.e., $\eta_{max}/\eta_0 \sim T_{gate}^{0.95}$, while **h** it appears to decrease with increased viscosity reduction, as $\eta_{max}/\eta_0 \sim (\eta_0/\eta_{min})^{-0.47}$.

To quantify the time-varying viscosity shown in Fig. 5b on [EcoRI], we evaluate the functional forms of $T_{gate}$, $\eta_{max}/\eta_0$, and $\eta_0/\eta_{min}$. As shown in Fig. 5d–f, both $T_{gate}$ and $\eta_{max}/\eta_0$ exhibit approximately power-law decrease with [EcoRI] with scaling exponents of ~0.47 and ~0.26, respectively. The change in large time viscosity with respect to the original viscosity, $\eta_0/\eta_{min}$, likewise exhibits approximate power-law dependence but with a positive and stronger scaling exponent of ~0.75.

To clarify the relationship between these observables, we also plot the relative viscosity amplification ($\eta_{max}/\eta_0$) and reduction ($\eta_0/\eta_{min}$) (Fig. 5g, h). We observe that the viscosity amplification ($\eta_{max}/\eta_0$) increases nearly linearly with gating time ($\eta_{max}/\eta_0 \sim T_{gate}^{0.95}$), whereas it decreases with increasing viscosity $\eta_{max}/\eta_0 \sim \eta_0/\eta_{min}^{-0.47}$ (Fig. 5g, h). Based on these relations, we expect the increase and subsequent decrease in viscosity to be independently tunable by varying the DNA concentration, length and stoichiometry of the linearising RE. Conversely, we anticipate that the gating time and degree of viscosity reduction will remain primarily dictated by the concentration of the multi-cutter RE and the average fragment length $\langle l_f \rangle$ it produces.

## Conclusions

Here, we study a class of polymeric fluids that are pushed out-of-equilibrium by dissipative topological alterations to their architecture. Specifically, we realise DNA fluids in which restriction enzymes cut DNA to trigger the relaxation of the system from one equilibrium state to another. As such, our fluids are en-route between two equilibrium states, during which they exhibit a wide range of time-varying rheological properties which depend on the parameters of our systems.

Moreover, we demonstrate that the initial and final states, relaxation kinetics, and rheology of the fluids can be precisely tuned by varying the DNA concentrations, lengths and topologies, as well as the concentration and type of REs. Importantly, we find that the link

between structural alterations and time-varying rheology is generally governed by the conformational size of the molecules and the ensuing entanglements.

Intriguingly, and in contrast with the common assumption that DNA digestion decreases the viscosity of DNA fluids, we show that linearisation of circular DNA increases the viscosity in a manner that is dictated by the time-varying degree of overlap between the molecules (Figs. 1–3). Conversely, the viscosity of entangled linear DNA undergoing fragmentation decreases at a rate that is universally tuned by the time-varying average fragment length, mediated by RE concentration (Fig. 4). Finally, we demonstrate that by coupling these two design strategies to creating non-equilibrium rheological behaviours we can create DNA fluids that exhibit non-monotonic and gated variations in viscosity (Fig. 5). In particular, we show how to achieve a short-time increase in viscosity followed by time-gated viscosity reduction, and how to tune the rheological features by varying the different system inputs, such as DNA concentration, length and RE type.

The topologically-driven time-dependent rheological properties we demonstrate here, along with the relationships that we elucidate between the architecture of the DNA constituents and their bulk rheological properties, may also facilitate future materials applications. Further, our work may shed new light on the viscoelasticity of entangled genomic material in cellular processes such as DNA repair, where transient DNA breaks are required. Finally, our strategy allows for the efficient and precise preparation of a wide variety of topological blends 'in one pot'. By slowing enzymatic digestion, we can generate widely different blend compositions in a single sample chamber, rather than preparing different blend formulations by hand, thereby saving sample quantity and avoiding concentration inconsistencies. This approach allows for remarkably high resolution in formulation space of polymer blends via in situ digestion.

## Methods

### DNA and restriction endonucleases

For data shown in Figs. 1 and 3, we use pYES2 DNA (5.9 kbp) prepared via replication of cloned plasmids in E. coli, followed by extraction, purification, and vacuum concentration[54,81]. pYES2 and the purification and characterisation methods we use have been thoroughly described and validated in the literature[38,39,44,54,81]. The resulting 13 mg/ml stock DNA solution, comprising 55% supercoiled circular topology and 45% nicked circular (ring) topology, is suspended in nanopure deionized (DI) water and stored at 4 °C. We quantify DNA concentration and topology via quantitative analytical gel electrophoresis as described below (see also Supplementary Fig. 3). We note that while the bulk nature of gel electrophoresis quantification limits its precision to within ~5−10%, we have previously shown that using single-molecule microfluidic stretching experiments to determine the fraction of each topology comprising our DNA solutions yield results that are comparable (within 2−3% variation) to those we measure via gel electrophoresis (further described in Supplementary Fig. 4 and ref. 39).

For the data shown in Fig. 4, we use $\lambda$-DNA (48.5 kilobasepairs (kbp)), supplied from New England Biolabs at 0.5 mg/ml in TE (10 mM Tris-HCl, 1 mM EDTA, pH 8) buffer (N3011), which we concentrate to 1 mg/ml via ethanol precipitation. We note that this construct has been used for decades as a model polymer system and its characterisation has been published extensively[46,70–72]. The day before a digestion experiment, a 100 μl DNA sample is incubated at 60 °C for 5 min and quenched with 1 μl of a 100 μM solution of 12 bp single-stranded DNA oligos purchased from IDT (GGGCGGCGACCT and AGGTCGCCGCCC, ~10:1 stochiometric ratio) to avoid hybridisation of $\lambda$-DNA due to complementary overhangs at the cos sites. We quantify the effect of RE digestion on the average number of cleaved fragments $\langle n_f \rangle$ and the corresponding average fragment length $\langle l_f \rangle$ as a function of digestion time $t_a$ by performing time-resolved gel electrophoresis (Fig. 4j) and analysing the band intensity following the procedures of ref. 80. In brief, we use a standard reference ladder ($\lambda$-HindIII digest) to create an interpolation function to map gel pixel to DNA length, then measure the intensity of each band and divide it by the expected length of the molecules in that band to determine their relative abundance. From the quantified relative fractions of chain fragments, we compute $\langle l_f \rangle$.

For data shown in Fig. 5 we use a 44 kbp DNA (IE241, gift from Aleksandre Japaridze) prepared via replication of cloned plasmids in Escherichia coli followed by extraction and purification as described above. Stock solutions of IE241 are composed of supercoiled and nicked circular topologies (Supplementary Fig. 6). All restriction enzymes (RE) are purchased from New England Biolabs and stored at −20 °C.

For all REs, if available, we use the high-fidelity (HF) versions of these enzymes to reduce non-specific star activity. The specific REs we use include: BamHI-HF, HindIII-HF, EcoRI-HF, ScaI-HF, HaeIII, XhoI and EcoRI-HF. See Supplementary Table 1 for more details.

### Digestion reactions

RE digestion reactions are prepared by adding 1/10th of the final reaction volume of 10x reaction buffer supplied by the RE manufacturer (typically NEB CutSmart buffer) and 0.1% Tween-20 to the DNA solution (diluted to the desired concentration in nanopure DI water). Following mixing of the solutions by pipetting with a wide-bore pipet tip (to avoid shearing DNA) or putting on a roller, the RE is added and mixed and the solution is incubated at RT for the duration of the digestion. For experiments shown in Figs. 1 and 3, we use pYES2 DNA concentrations of $c = 1.5$−6 mg/ml, corresponding to ~2$c^*$−8$c^*$ of the undigested solution of circular molecules and ~9.5$c^*$−38$c^*$ for linear pYES2 DNA. We also vary the stoichiometric ratios of enzyme units to DNA mass from 0.01 to 0.5 U/μg. For experiments shown in Fig. 4, we perform digestions at $\lambda$-DNA concentration $c = 1$ mg/ml

(~20$c^{*69}$) and stoichiometric RE:DNA ratios of 0.27−5.5 U/μg. For Fig. 5 experiments, we use a concentration of $c = 1.4$ mg/ml IE241 plasmid DNA, corresponding to ~12$c^*$, and stoichiometries of 0.84 U/μg XhoI and 0−0.84 U/μg of EcoRI-HF.

### Time-resolved gel electrophoresis

To characterise the rate at which REs digest DNA under the different conditions we use direct current agarose gel electrophoresis to separate the different topologies (linear, supercoiled, ring) and lengths of DNA. Specifically, we prepare a 40 μl digestion reaction as described above and incubate at RT for 4−12 h. Every 5−10 min during the digestion we remove a 1 μl aliquot from the reaction and quench it with TE buffer and gel loading dye. We load 50 ng of DNA from each 'kinetic aliquot' onto a 1% agarose gel prepared with TAE buffer. We run the gel at 5 V/cm for 2.5 h, allowing for separation of the DNA into distinct bands corresponding to supercoiled, ring, and linear DNA of varying lengths. We use the standard $\lambda$-HindIII molecular marker (M) to calibrate the gel and determine topology, length and concentration of the different DNA bands in our samples. We use a Life Technologies E-Gel Imager to image DNA bands on the gels and use Gel Quant Express software to perform image intensity analysis to determine the relative concentrations of each band.

### Time-resolved microrheology

For microrheology experiments, we mix into the digestion reaction a trace amount of polystyrene microspheres (Polysciences), of diameter $a = 1$ μm, coated with Alexa-488-BSA to inhibit binding interactions with the DNA and visualise beads during measurements. We load the sample into a 100-μm thick sample chamber comprising a microscope slide, 100 μm layer of double-sided tape, and a glass coverslip to accommodate a ~10 μl sample. The chambers are sealed with epoxy to avoid evaporation over the course of the several hour experiments.

We perform experiments using an Olympus IX73 microscope with a 40x objective or a Nikon Eclipse Ts2 with 60x objective and Orca Flash 4.0 CMOS camera (Hamamatsu). We record time-series of diffusing beads starting ~5 min after adding the RE to the sample (enough time for the epoxy to dry and to load the sample onto the microscope) and every subsequent 10 min for the duration of the experiment. Time-series are collected for 120 s at 20 fps on a 1024 x 1024 field of view (imaging ~100 particles per frame), resulting in ~500 tracks per time-series.

Temperature is precisely maintained at 25 °C in the microscopy lab where experiments are performed. We do not use a temperature-controlled unit on the microscope, but the low light intensity needed to image commercial fluorescent-labelled microspheres, ensures negligible local heating, as corroborated by our steady-state control experiments that show no signs of local heating.

We use TrackPy[82] and custom-written particle-tracking codes (in Python and C++) to extract the trajectories of the diffusing beads and measure the time-averaged MSDs of the diffusing particles as a function of lag time $t$. We note that while the tracked trajectories are in 2D, because the samples are isotropic, we average the $x$ and $y$ direction as if they were independent walks, such that all MSDs shown and used to determine viscosities, are determined from the average of the MSDs in the $x$ and $y$ directions.

We compute diffusion coefficients $D$ via linear fits to the MSDs according to MSD = $(2d)Dt$ (with $d = 1$, because the $x$ and $y$ directions from 2D tracking are averaged together). From $D$, we compute the zero-shear viscosity $\eta$ using the Stokes−Einstein equation $\eta = k_B T / (3\pi D a)$ with $a$ the diameter of the particles. Strictly speaking, this approach is only valid for Newtonian fluids that exhibit purely viscous behaviour, and the highest DNA concentrations we examine exhibit modest viscoelasticity and subdiffusion at short lag times for certain $t_a$ (Figs. 1m and 4d). However, for these cases we restrict our analysis

to the large-time terminal regime in which the MSD scales linearly in time and the complex viscosity $\eta^*(\omega)$ is independent of frequency $\omega$.

While our analysis ignores bead motion in the $z$ direction, we only track beads that are in focus, so their $z$ movement (while tracking in $x$ and $y$) is limited to roughly their radius (500 nm), which correlates to an MSD of order $0.1\,\mu m^2$. This limit is below the large lag time MSDs that we use to determine diffusion coefficients and viscosities. It is still possible that MSDs are systematically slightly underestimated by ignoring $z$ motion, but this effect should have minimal impact on the metrics we extract from the MSDs and the trends we present. Finally, because the beads we track are ~2–10x larger than $R_g$ for any of the DNA molecules or fragments we study, any conformational changes to the DNA that occur in $z$ should be captured in the 2D traces of the beads.

Error bars shown in Figs. 1–4 are determined by computing MSDs and resulting viscosities from five random subsets of each dataset and computing the standard error across the subsets.

We note that while our system is out-of-equilibrium, we tune the digestion rates to be several orders of magnitude slower than the longest polymer relaxation times of the fluids as well as the video acquisition time. Specifically, the fastest time at which full digestion occurs is $\tau_c \simeq 30\,min$ (see Figs. 1 and 4) compared to polymer relaxation time of $T_d \simeq 2\,s$ and data acquisition time of $120\,s$. As such, the number of cleavages within one relaxation time is $\chi \equiv T_d/\tau_c \lesssim 10^{-3}$ and is $\lesssim 0.05$ within one acquisition time. Thus, we can consider our system to be in quasi-steady-state with respect to the architectural degrees of freedom during each of our time-resolved measurements, allowing for unambiguous determination of diffusion coefficients and viscosity.

### Generalised Stokes–Einstein relation

In order to obtain the frequency-dependent complex viscosity and moduli of the fluids from MSD($t$) curves we use the generalised Stokes–Einstein relation, adopting the method by Mason[55], which we implement in custom-written Mathematica code. We first interpolate the MSD with a 2nd-order interpolation curve $f(t)$, then compute the time-dependent exponent as

$$\alpha(t) = \frac{d\log f(t)}{d\log t}, \tag{3}$$

which can be expressed in the frequency domain as

$$\alpha(\omega) = \frac{d\log f(t)}{d\log t}\Big|_{t=1/\omega}, \tag{4}$$

and use the semi-analytical formula for the absolute value of the complex modulus[55]

$$|G(\omega)^*| = \frac{k_B T}{\pi a(i\omega)\mathrm{MSD}(1/\omega)\Gamma[1+\alpha(\omega)]}. \tag{5}$$

From this expression, we can obtain

$$G'(\omega) = |G^*(\omega)|\cos\pi\alpha(\omega)/2, \tag{6}$$

$$G''(\omega) = |G^*(\omega)|\sin\pi\alpha(\omega)/2, \tag{7}$$

$$\eta^*(\omega) = \frac{G^*(\omega)}{i\omega}. \tag{8}$$

### Brownian dynamics simulations

As detailed in Supplementary Note 1 and in refs. 12,65, we model DNA as a twistable elastic chain. Potentials (detailed in the Supplementary

Note 1) are set to account for excluded volume, bonding and chain stiffness in line with those known for DNA. The simulations are performed at fixed monomer density $\rho\sigma_b^3 = 0.08$ and $\rho\sigma_b^3 = 0.006$, equivalent to ~39 mg/ml and 3 mg/ml of DNA ($\sigma_b = 2.5\,nm = 7.35\,bp$ is the typical size of a bead). The equations of motion for each bead are evolved and coupled to a heat bath, which provides noise and friction. The equation of motion for each Cartesian component is

$$m_a\partial_{tt}r_a = -\nabla U_a - \gamma_a\partial_t r_a + \sqrt{2k_B T\gamma_a}\eta_a(t), \tag{9}$$

where $m_a$ and $\gamma_a$ are the mass and the friction coefficient of bead $a$, and $\eta_a$ is its stochastic noise vector satisfying the fluctuation-dissipation theorem. $U$ is the sum of the energy fields described. The simulations are performed in LAMMPS[83] with $m = \gamma = k_B = T = 1$ and using a velocity-Verlet algorithm with integration time step $\Delta t = 0.002$ $\tau_B$, where $\tau_B = \gamma\sigma^2/k_B T \simeq 0.03\,\mu s$ (using $\gamma = 3\pi\eta_{water}\sigma$ with $\eta_{water} = 1\,cP$ and $\sigma = 2.5\,nm$) is the Brownian time. To mimic different stages of DNA digestion by restriction enzymes with single restriction sites we remove a single bead from a different fraction $f$ of rings together with the angles and dihedrals in which it is involved. Subsequently, we set the dihedral constant of all the dihedrals belonging to the fraction $f$ of rings to $k_\psi = 0k_B T$ mimicking fully relaxed linear segments of DNA. This implementation assumes, as suggested by recent simulations, that twist diffuses much faster than writhe[66].

## Data availability

The data that support the findings of this study are available from the corresponding authors upon request. The source data underlying Figs. 1–5 are provided as a Source Data file. A reporting summary for this Article is available as a Supplementary Information file. Source data are provided with this paper.

## Code availability

The codes to perform the supercoiled DNA simulations can be found at https://git.ecdf.ed.ac.uk/taplab/supercoiledplasmids.

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

## Acknowledgements

D.M. acknowledges enlightening discussions with Rick Morgan (NEB) and Mark Szczelkun (Bristol) and thanks Aleksandre Japaridze (TUDelft) for providing IE241 plasmids. D.M. acknowledges the Royal Society for support through a University Research Fellowship and the European Research Council (ERC) for providing funding under the European Union's Horizon 2020 research and innovation programme (grant agreement no. 947918, TAP). R.M.R.-A. acknowledges the US Air Force Office of Scientific Research (Grant no. AFOSR- FA9550-17-1-0249) for support.

## Author contributions

D.M. and R.M.R.A. designed the research, supervised experiments, interpreted data and wrote the paper. D.M., P.N., S.W., N.C. conducted experiments and analysed data. All authors contributed to writing the paper.

## Competing interests

The authors declare no competing interests.
