## [Peer Review File · Nature Communications]

Topological digestion drives time-varying rheology of entangled DNA fluidsReviewers' comments:

Reviewer #1 (Remarks to the Author):

In this work the authors study the microrheology of complex fluids of DNA molecules undergoing structural changes due to endonuclease reactions (ERs). The authors present contrasting rheological behaviour of these systems, which they attribute to the intrinsic non-equilibrium nature of the system they study. The authors demonstrate that by varying the enzyme type and concentration, the associated rheological properties can be altered, with two contrasting examples: (a) entangled linear DNA undergoing digestion shows a drop in viscosity over time, and (ii) circular DNA undergoing digestion displays an increase in viscosity. The authors attribute the differences to the underlying topological transformation in the DNA structure. Overall, the paper is well written, though recent works on topological microfluidics (both active and passive) could have found mention in the references. The authors have proposed DNA-based liquids as complex fluids which exhibit time-varying rheological attributes, mediated specifically by the ER-driven topological changes. The manuscript is well-written and timely, considering the recent focus on topological microflows, and active anisotropic complex fluids. However, despite this exciting premise, it suffers from the following major weaknesses (listed below). The Referee missed the broader context of the results presented, since links (specifically to the potential applications) proposed in the final sections of the paper are largely speculative. In summary, I am not able to recommend publication of this work in Nature Communications at this stage, since the novelty of the work (fundamental, technical and applied) is ambiguous, and the results, to an extent, are incremental. I would be happy to see a revised version (should there be a chance for that) which reports the topological changes quantitatively, both in space and time, and delineates a clear coupling between topology and the observed time-dependent rheology claimed in this paper.

Major concerns:

1) The authors claim that the time-dependent rheology emerges as a consequence of the topological changes in the DNA structure, however quantitative data supporting the same are lacking. Topology-governed rheology has been implied as a central novelty of this work, however changes in DNA topology in the experiments were not substantiated through data. While topological changes - as a consequence of ER - can be inferred, so are the changes in the geometry and radius of the gyration of the DNA molecules. How do the authors distinguish these confounding factors in their study? In the current version, it is not adequately clear how (and why) the authors selectively choose topology, though entanglement and changes in gyration radius have long been implicated for wide-ranging rheological manifestations of DNA liquids.

2) The authors have, in their previous works (some of which are cited here), looked in to the steady-state rheological implications of ER-driven DNA structures. In this sense the key difference in this work is the time-series data of the rheology, as the ER progresses. It thus becomes critical that the authors identify (and justify) the temporal resolution of the acquired data. In the limit of large timescales, the rheological manifestations would reflect the properties of the steady state DNA liquids (a topic the authors have previously reported on). Thus, novelty of this work shifts toward the technical details of carrying out the time-series measurements (as against steady-state measurements in their previous works), and not as much on the novelty of the rheological data presented. It could thus be relevant to compare the timescales of structural changes with those of the rheological changes. In this context

the temperature at which the experiments were conducted will be crucial, so a note on the temperature stability (and controllability) will be a valuable addition.

3) Although the authors have presented rheological data over time, the temporal evolution of the properties is not linked appropriately with the corresponding topological changes. In this regard, the simulation data is sparse, and could have provided valuable insights. The section on the MD simulations reads considerably truncated, as only a very small part of the experimental data is captured through the simulation results. The angle of programmable rheology could have been extensively captured through the simulations, though the authors have not provided any insight therein. Finally, the simulations could come handy in establishing the topology-rheology link, an angle which has been left out despite its central premise in this work.

4) Since the ER is expected to progressively alter the structure of the DNA complex fluid, how do the authors rationalize the rheology data with the partial transformation of the structures (at any intermediate time point in the ER)? For instance, can the authors fit the rheology data with the “effective structure” of the DNA fluid at any instant of experiment? Importantly, what would be the corresponding topological metric of the system? How do the authors link DNA topology (which is a local property at any instance of ER) and the observed rheology (which, for the relative scales, may be a bulk property) at any time point of the ER. Consequently, have the authors observed spatial variations in their microrheology experiments (at any intermediate time point), or do the authors observe spatially homogeneous rheology?

5) Structural changes due to ERs are expected to occur in 3D, although the authors quantify particle mobility (in the microrheology experiments) using 1D mean squared displacement. The authors should justify this difference in dimensionality, and comment on its impact in the interpretation of the presented data.

6) The title and motivation of the manuscript limits the scope of the manuscript. The authors should bring out the significance of their results clearly (possibly along the above suggested points), which together with additional experiment and simulations, could enhance the novelty of the presented results.

Minor comments:

- 1) The authors should spell out RE at its first appearance in the text
- 2) k in characteristic cutting time should be defined at its first point of appearance
- 3) Figure numbers are styled/formatted inconsistently, the authors could make these consistent

Reviewer #2 (Remarks to the Author):

The paper studies how topological and architectural alterations of DNA solutions driven by endonuclease reactions control the rheology of the fluids. The key findings of the paper include: 1. entangled linear DNA solutions undergoing restriction enzyme digestion display a marked decrease in viscosity; 2. circular DNA solutions undergoing restriction enzyme digestion display a monotonic increase in viscosity; 3. combining restriction enzyme of different cutting frequencies, the viscosity of circular DNA solutions first increases due to entanglement of linearized chains and then decreases due to further fragmentation of the

linear chains. The experiments were explained by scaling analysis and supported by molecular dynamics simulations.

The results are solid and the experiments/analysis were done in an expert manner. I found the paper very interesting while reading the abstract/introduction and was looking forward to see results on "a wide range of non-equilibrium rheological behaviours that can be programmed". But in the end, the authors only briefly touched on the programmable dynamics (Fig. 5). The general principle of using enzymatic reactions to control molecular topology and flow properties of DNA solutions dates back to 2010 (PMCID: PMC2941032), and recent publications from the senior author's lab have reported very interesting findings along this line. Although restriction enzymes are now used instead of topoisomerase, the general principle is similar. Also restriction enzyme digestion is known by molecular biologists to cause reduction of viscosity; for example, see this article PMCID: PMC4948299 found by a quick search. Results in Fig. 5 are straightforward. A more creative and systematic design of programmable dynamics of controlled fluid rheology (including viscoelasticity, another important aspect overlooked by the authors) would substantially strengthen the paper. I really like the aspirational discussion in the paper, but results presented in the current paper seems not yet ready to serve the aim.

Below are some comments for the author's consideration:

1. The authors tried to make connection to active matter. There is no local free energy input in the experimental system studied in this paper, and although the system can just be called as out-of-equilibrium, the dynamics is driven by free energy minimization like in all regular chemical reactions. The connection to active matter sounds vague, unnecessary, and even confusing, especially in view of the many existing studies on self-propelled polymers.
2. The authors could elaborate how viscoelasticity of the fluids were affected or controlled.
3. Certain points discussed in Conclusion are a bit overstretched given the existing results.
4. The authors called the experimental system as "topologically-active fluids". In the first part of the results (cutting linear chains shorter), I do not see the change of molecular topology.
5. The authors use gel electrophoresis to determine the topology forms of DNA molecules. Could the authors provide any microscopy examination or references to help readers understand the molecular form of the bands?
6. A more detailed version of scaling analysis should be provided somewhere, such as SI.
7. The authors wrote "indeed, if this was the case we would expect a non-monotonic behaviour in viscosity with a maximum at low ring contaminants within a solution of linear chains [31] (expected at large digestion time) due to threading". Why a maximum at low ring contaminants? According to Ref. 29, ring threading will further increase viscosity, then why would the authors expect a non-monotonic behaviour in viscosity?
8. The authors described XhoI as 2-cutter and EcoRI as 10-cutter because the RE cut twice or 10 times on the plasmid. This is really confusing, because the numbers in n-cutter refers to the length of recognition site of an RE.
9. Fig. S3: Please show the full ladder markers to help readers better tell the mobility of

different bands. Fig. S4: Please indicate what each band is.

10. Fig. S5: Ref 3 did not show data for 0.5 g/ml DNA concentration.

Reviewer #3 (Remarks to the Author):

Dear Editor,

This manuscript presents an elegant and timely series of experiments and simulations that's explore topologically complex blends of ring and linear DNA as a rheologically programmable active fluid system. The authors use natural enzymes to actively transform DNA dispersity and topology and demonstrate that this can be engineered to significantly increase or decrease fluid viscosity in a controlled way. The authors present simple and compelling arguments to explain their linear chain digestion data, and they provide complementary simulations for the ring-digestion case where no simple arguments are at hand.

While I defer analysis of the experimental details to those more qualified, I have examined the simulation methods and find no issue with them that would invalidate the authors' core conclusions. However, I do believe that their discussion of their simulation methodology requires some additional supplemental details – as I discuss below.

In addition, the authors have kept their discussions very focused on DNA systems, but ring and ring/linear blend rheology's have also been a popular research topic in non-DNA systems – the Journal of Rheology just released a special issue with many studies in this area. To emphasize the broad impact of their study, I'd encourage the authors to add some additional context and contrast between their observations and those seen in non-supercoiled polymer blends.

I recommend this study for publication in Nature Communications provided the authors respond to these comments as well as those below.

###Comments Regarding the Main Text

#Terminology “rheopecty” – In the abstract the term rheopecty is used. In the conclusion, the term anti-thixotropy is used instead. I'd recommend unifying the language used for this increasing low-rate viscosity.

#Initialisms – The authors use the initialism RE before defining a couple paragraphs later.

#Linearizing Chains in Simulation — The main text very briefly describes the molecular model for digestion as “linearizing chains,” but I think this is a bit too ambiguous. I think it would be appropriate for the authors to include a few more details in the main text that indicate what is meant by linearizing (ring opening + removal of torsional stiffness).

###Comments Regarding Modeling Methods In the Supplement

#Patch modeling -Please specify how “patches” are modelled in your bead-spring polymer

model. Do your individual beads track an internal vector orientation, or are your patches modeled with additional beads affixed to the backbone beads with rigid-body constraints? Patches and rotational inertia - Are the patches decorating the beads given mass to produce a moment of inertia along the backbone? If so, please specify. If not, do the authors observe any unusual behavior due to an absence of "torsional inertia."

#FENE Bond parameters -The choice of FENE bond potential parameters is just slightly different than the values typically used - $k=30\epsilon/\sigma^2$ and $R_0=1.5\sigma$. Can the authors include a sentence in their methods rationalizing this slightly different choice? Is it to shift the average bond-length closer to 1σ ?

Removing Torsion in Digested DNA – The authors use a torsional stiffness with a controlled pitch to drive ring DNA chains to twist. This torsion potential is removed when they want to study uncoiled DNA.

I may be naïve to the mechanism of supercoiling, but I would appreciate the authors explaining why they completely zero-out the torsional stiffness when they model ring digestion. I understand the goal is to create a fraction of uncoiled DNA, but can the authors comment on their model versus the real system?

I'd imagine that the real system supercoils due to stored torsion in the ring backbone which has a torsional stiffness (twists along the chain) that cannot unwind due to the ring topology. Cutting the ring allows this stored torsion to unwind and shuts off the twist-bend instability. Is this a correct understanding?

If so, then wouldn't the linear DNA still have a torsional stiffness, and could it have some influence on the conformations of the linear DNA? Have the authors examined the conformational statistics of the linear DNA with and without a torsional stiffness turned on? Aside from efficiency, is there any reason the authors couldn't model supercoiled DNA by pre-twisting their rings and allowing them to unwind?

Response to Reviewer Comments

Reviewer comments are reproduced in green font

Excerpts from the revised manuscript are in blue font. References are not included in the excerpts (often replaced with [..]) but can be found in the manuscript.

Reviewers' comments:

Reviewer #1 (Remarks to the Author):

In this work the authors study the microrheology of complex fluids of DNA molecules undergoing structural changes due to endonuclease reactions (ERs). The authors present contrasting rheological behaviour of these systems, which they attribute to the intrinsic non-equilibrium nature of the system they study. The authors demonstrate that by varying the enzyme type and concentration, the associated rheological properties can be altered, with two contrasting examples: (a) entangled linear DNA undergoing digestion shows a drop in viscosity over time, and (ii) circular DNA undergoing digestion displays an increase in viscosity. The authors attribute the differences to the underlying topological transformation in the DNA structure. Overall, the paper is well written, though recent works on topological microfluidics (both active and passive) could have found mention in the references. The authors have proposed DNA-based liquids as complex fluids which exhibit time-varying rheological attributes, mediated specifically by the ER-driven topological changes.

We thank the referee for finding our work well-written and suggesting referencing topological microfluidics to better contextualize our work. As suggested, we have mentioned these studies and added corresponding references to the introduction.

The manuscript is well-written and timely, considering the recent focus on topological micro-flows, and active anisotropic complex fluids. However, despite this exciting premise, it suffers from the following major weaknesses (listed below). The Referee missed the broader context of the results presented, since links (specifically to the potential applications) proposed in the final sections of the paper are largely speculative. In summary, I am not able to recommend publication of this work in Nature Communications at this stage, since the novelty of the work (fundamental, technical and applied) is ambiguous, and the results, to an extent, are incremental. I would be happy to see a revised version (should there be a chance for that) which reports the topological changes quantitatively, both in space and time, and delineates a clear coupling between topology and the observed time-dependent rheology claimed in this paper.

We thank the reviewer for finding our manuscript 'well-written' and 'timely' with an 'exciting premise'. We also thank the reviewer for welcoming a revised manuscript that addresses their comments. We have substantially revised and expanded our manuscript to more clearly describe the novelty of the platform and results, and to more quantitatively and exhaustively couple the DNA topology to the rheology we present. In our responses to the specific comments below, we expand on these revisions and additions, which include new experiments and simulations and substantially expanded data analysis and interpretation. We trust with these significant changes that the reviewer will find our work of interest to the broad interdisciplinary readership of Nature Communications.

Major concerns:

1) The authors claim that the time-dependent rheology emerges as a consequence of the topological changes in the DNA structure, however quantitative data supporting the same are lacking.

Topology-governed rheology has been implied as a central novelty of this work, however changes in DNA topology in the experiments were not substantiated through data. While topological changes - as a consequence of ER - can be inferred, so are the changes in the geometry and radius of the gyration of the DNA molecules. How do the authors distinguishing these confounding factors in their study? In the current version, it is not adequately clear how (and why) the authors selectively choose topology, though entanglement and changes in gyration radius have long been implicated for wide-ranging rheological manifestations of DNA liquids.

We thank the reviewer for raising this point and requesting more clarity. We use quantitative image analysis of gel electrophoresis experiments to unambiguously determine the size and topology of the DNA at each time point at which we measure the rheology for the full duration of each digestion experiment. In the original manuscript we did not describe this analysis in much detail and included most of the gel electrophoresis images in the SI. We now describe this analysis in the main text and in the SI and include an explanation figure in SI (**SI Fig S4**). We also explicitly relate the topology and length of the DNA to the measured viscosity. For example, in **Fig 1**, in which we observe rheopectic behavior due to linearization of circular plasmids, we now show several sample gel electrophoresis images and plot the fraction of linear DNA as a function of time, as determined via gel electrophoresis analysis. More importantly, to directly correlate rheology to topology, we plot the normalized viscosity η/η_0 as a function of linear DNA fraction ϕ_L . In **Fig 3**, we relate the average radius of gyration $\langle R_g \rangle$ (averaged over all topologies) to the linear fraction and measured diffusion coefficient D for experiments *and* Brownian Dynamics simulations. Finally, in **Fig 4**, in which we present thixotropic behavior of entangled linear DNA undergoing fragmentation, we show the correlation between η/η_0 and mean fragment length $\langle l_f \rangle$, determined via gel electrophoresis and non-equilibrium living polymer theory. We expand on these new analyses below.

To shed light on the physics underlying the correlation between topology and rheology, we use Brownian Dynamics (BD) simulations and develop non-equilibrium living polymer theory. Our BD simulations, which recapitulate our **Fig 1** experiments (as demonstrated in **Figs 2, 3**), reveal that the conversion from circular to linear topology results in an increase in polymer coil size (both $\langle R_g \rangle$ and root-mean end-to-end distance $\langle R_{ee} \rangle$), and it is this increased size that leads to the reduction in diffusion coefficient. This effect is more apparent at higher polymer concentrations, indicating that the increasing polymer coil overlap (quantified by the overlap concentration c^*) resulting from changing topology contributes to the topology-driven rheology.

Based on these results, we use our quantitative gel analysis and previously reported values and relations for R_g of supercoiled, ring and linear DNA to compute a time-dependent $\langle R_g \rangle$ and an effective overlap concentration c^* for each topological state of the DNA fluid (i.e., each experimental time point), as we now show in **Figs 3 and S5** and *explain in the revised manuscript as follows*:

As our simulations indicate that the changing conformational size of the polymers, as well as the degree of entanglement (i.e., c), are driving factors in the rheopecty we measure experimentally, we next compute $\langle R_g \rangle$ as a function of time for the data shown in Fig 1. We use previously reported R_g values and relations $R_{g,L}/R_{g,R} \simeq 1.6$ and $R_{g,L}/R_{g,SC} \simeq 2.1$, along with ϕ_L quantified from gel electrophoresis analysis to compute $\langle R_g \rangle$, averaged over all topologies, as a function of digestion time t_a and linearised fraction ϕ_L . As shown in SI Fig S5, $\langle R_g \rangle$ increases with ϕ_L , similar to simulations. The impact of increasing $\langle R_g \rangle$ on the dynamics in experiments and simulations is captured in Fig 3b, in which we plot D/D_0 as a function of $\langle R_g \rangle$. As shown, D/D_0 decreases with increasing $\langle R_g \rangle$, with remarkably similar function dependence for experiments and simulations. Notably, in both cases, the

more rapid drop in D/D_0 for higher DNA concentrations is stark. Moreover, in all cases, the dependence is stronger than the $D \sim R^{-1}$ Stokes dependence expected in the dilute case for both simple spheres as well as ring, supercoiled and linear polymers, once again pointing to the impact of polymer overlap in the rheology.

In semidilute and concentrated solutions, changing R_g directly alters the degree of overlap via the relation $c^* = (3/4\pi)(M/N_A)/(R_g^3)$ where c^* is the polymer overlap concentration. To determine the role of topology-driven changes in polymer overlap, owing to the changing size of the polymers, we plot the normalized viscosity η/η_0 against c/c^* . Note that c/c^* will decrease if c decreases or if R_g increases (as it does during digestion). To compute c^* during digestion, in which we have a mix of different topologies – each with a different R_g – we start with the conventional expression for c^* , derived by equating the solution volume (m/c^* where m is total mass) to the total volume the molecules comprise, i.e. the total number of molecules $N = mN_A/M$ multiplied by the volume per molecule $V_m = (4\pi/3)R_g^3$. We then consider that each component contributes separately to the total volume the molecules fill in solution: $N_L V_{m,L} + N_R V_{m,R} + N_S V_{m,S} = (4\pi/3)(\phi_L R_{g,L}^3 + \phi_R R_{g,R}^3 + \phi_S R_{g,S}^3)$. The resulting expression is then: $c^* = (3/4\pi)(M/N_A)/(\phi_L R_{g,L}^3 + \phi_R R_{g,R}^3 + \phi_S R_{g,S}^3)$.

As shown in Fig 3c, our normalized viscosity data η/η_0 all collapse onto a master curve when plotted against c/c^* , with a functional form that can be described by $\eta/\eta_0 \sim (c/c^*)^\gamma$ where $\gamma \approx 0.5$ and $\gamma \approx 1.75$ at low and high c/c^* values, respectively. The crossover in scaling takes place at $c \approx 6c^*$ which we previously showed to be at the onset of entanglement dynamics for DNA solutions [..]. The exponents are in agreement with those reported in single-molecule tracking studies measuring the diffusion of ring and linear DNA in semidilute ($c \approx c^*$) and entangled ($c > 6c^*$) DNA solutions, respectively [..]. Specifically, these studies showed that $D \sim c^{-1.75}$ for 11, 25 and 45 kbp ring and linear DNA at concentrations above $6c^*$, in line with reptation model predictions. Below $6c^*$, diffusion coefficients followed $D \sim c^{-0.5}$ scaling, in accord with Rouse model predictions. These exponents, compatible with the behaviour of semidilute and entangled flexible polymers, differ from those measured via bulk rheology for semiflexible polymers and for theta solvent conditions [..].

Taken together, our experiments and simulations demonstrate that the RE-driven rheology we observe arises from the conversion of circular to linear polymers that increases the average conformational size of the polymers, which, in turn, increases the degree to which polymers are overlapping and entangled (see Fig. 3d). The effect of topology is more dramatic in the entangled regime in which diffusive processes are more strongly dependent on concentration. As we describe above, our results shown in Fig 1 suggest that threading may contribute to the dynamics we measure, in which case we expect $D \sim c^{-3}$ rather than $D \sim c^{-1.75}$, as previously reported for ring-linear DNA blends in which constraint release from threading dominated the diffusivity. Indeed, for the highest concentrations ($c/c^* > 10$) the scaling of η/η_0 is steeper than $\gamma \approx 1.75$ and more closely follows $\gamma \approx 3$, indicating contributions from threading.

To directly couple DNA structure to rheology in the experiments in which we digest linear DNA into smaller fragments to produce thixotropic behavior (**Fig 4**), we have performed quantitative gel electrophoresis to measure the average length of the fragments $\langle l_f \rangle$ as a function of digestion time (**Fig 4**). We have also extracted $\langle l_f \rangle$ from fits of our data to our non-equilibrium living polymer model. Plotting η/η_0 as a function of $\langle l_f \rangle$ indeed confirms that thixotropy is directly correlated with the average filament length. To understand the functional form of this dependence and the underlying physics we turn to our extension of living polymer theory that fits our data remarkably well. From fits

of our data to this model, we extract cutting rates κ which we transform into average filament length $\langle l_f \rangle$ via $\langle l_f \rangle = L/(1 + \kappa t_a)$ where $L = 48.5$ kbp is the initial DNA length. Our plot of η/η_0 versus $\langle l_f \rangle$ demonstrates agreement with theoretically predicted functional dependence of viscosity on fragment length - unambiguously correlating thixotropy to decreasing filament length.

The reviewer is correct that changing the DNA topology indeed changes the conformational size as well as the nature and degree of entanglements, *as we describe above and in the text*, which may impact the non-equilibrium rheology we measure. Our new analysis dissects these confounding effects by showing that it is principally the degree of polymer overlap that programs the rheology. While this increase may be achieved by other means than varying the topology, such as buffer-mediated polymer swelling or compression of microfluidic chambers, we also show evidence of threading of ring polymers by linear chains that produce noisier and more heterogeneous dynamics that would not be present using alternative approaches to increasing overlap.

Our specific choice to focus on topology is motivated by the emergent behavior reported for topological blends of DNA and other polymers, and the still-debated nature of entanglements in solutions of dense circular DNA. For example, the differing impacts of ring-linear and ring-ring threading on the rheology of entangled rings and ring-linear blends is still a controversial topic. Moreover, there is a paucity of knowledge on the rheology of dense solutions of supercoiled polymers and how they interact with topologically distinct polymers, including the extent to which supercoiled polymers can participate in threading. Finally, one of the most elegant points that we aim to make in this paper is that simply by cutting a polymer at a single point - keeping the mass concentration of the DNA and the environmental conditions unchanged - we are able to alter the degree of overlap, the conformational size, and nature of inter-polymer interactions and entanglements.

We have added the following text to the introduction to highlight these points:

In the materials and engineering communities, polymer topology has long been appreciated for its ability to confer novel rheological properties to polymeric fluids and blends that can be tuned for commercial and industrial use [..]. In particular, end-closure of linear polymers (creating open ring, knotted and linked constructs) and breakage and fragmentation of circular polymers (resulting in linear chains) and their roles in the rheology and dynamics of entangled polymer systems is a vibrant and widely-investigated topic of research [..]. While the dynamics of entangled linear polymers are well described by the reptation model developed by de Gennes, Doi and Edwards [..], the extension of this model to circular polymers, with no free ends, is not straightforward. The extent to which ring polymers form entanglements, and the relaxation modes and conformations available to ring polymers remain topics of fervent debate [..].

Moreover, in blends of polymers of distinct topologies, such as ring-linear blends, the role of polymer threading and other topological interactions can lead to emergent rheological and dynamical properties such as increased viscosity, suppressed relaxation, and heterogeneous transport modes [..], compared to monodisperse systems of rings or linear chains. For example, solutions of concentrated ring polymers exhibit lower viscosity than their linear counterparts [..], while blends of ring and linear polymers exhibit higher viscosity than pure linear chains [..] and display unique behaviours under extensional stress [..].

Far less understood are the rheological properties of supercoiled polymers, for which DNA is an archetypal example [..]. Previous microrheology studies on semidilute blends of ring and supercoiled DNA have shown that blends exhibit entanglement dynamics at concentrations well below that needed for monodisperse systems of ring or linear polymers to exhibit similar viscoelastic properties [..]. At the

same time, simulations of dense supercoiled and ring polymers have shown that supercoiled DNA molecules exhibit faster diffusion and more swollen conformations than their ring counterparts [..].

The rich and surprising behavior of topologically-distinct polymeric fluids, along with the principal role that topology plays in dictating the conformational size and structure of the comprising polymers, as well as the nature of the interactions between them, inspired us to exploit polymer topology as a route to functionalize polymeric materials with spatiotemporally varying rheological properties. We use entangled DNA as our seminal 'topologically-active' polymeric material as DNA has been extensively employed as a model system to study polymer physics [..]. Further, DNA is particularly well-suited to study the role of topology on the rheology and dynamics of polymeric fluids as it naturally occurs in supercoiled, ring and linear conformations [..]. Finally, conversions from one DNA topology to another, via enzymatic reactions, play critical roles in myriad cellular processes such as replication and repair [..].

2) The authors have, in their previous works (some of which are cited here), looked into the steady-state rheological implications of ER-driven DNA structures. In this sense the key difference in this work is the time-series data of the rheology, as the ER progresses. It thus becomes critical that the authors identify (and justify) the temporal resolution of the acquired data. In the limit of large timescales, the rheological manifestations would reflect the properties of the steady state DNA liquids (a topic the authors have previously reported on). Thus, novelty of this work shifts toward the technical details of carrying out the time-series measurements (as against steady-state measurements in their previous works), and not as much on the novelty of the rheological data presented. It could thus be relevant to compare the timescales of structural changes with those of the rheological changes. In this context the temperature at which the experiments were conducted will be crucial, so a note on the temperature stability (and controllability) will be a valuable addition.

We thank the reviewer for raising these important questions. We first address the technical questions. As we describe in the main text, we collect data every 10 minutes over the course of 4 hours of RE digestion. Each time-series is captured at 20 fps for a duration of 2 mins, and the longest polymer relaxation timescale is of order 2 s. Temperature is precisely maintained at 25°C in the microscopy lab where experiments are performed. We do not use a temperature-controlled unit on the microscope, but the low light intensity needed to image commercial fluorescent-labeled microspheres, ensures negligible local heating, as corroborated by our steady-state control experiments that show no signs of local heating (see SI Fig S2). We have added this information to the Results and Methods section of the revised manuscript.

The data in **Figs 1, 3 and 4** show that the topological conversion time is on the order of 10s of minutes, so for each measurement point we can treat the system as in a quasi-steady state to facilitate interpretation of MSDs and their relation to viscosity. In this way, we can indeed compare our results to previous steady-state results when and if they exist (they often do not). However, the time-dependence of the rheology we present is from out-of-equilibrium topological conversion over time, and indeed reveals robust and programmable rheopecty and thixotropy. Capturing such time-dependence – far from incremental and not possible in steady-state measurements – is essential for the design of drug delivery systems or tissue regeneration scaffolds with time-dependent rheology.

We have added the following text to the manuscript to clarify:

We note that while our system is out-of-equilibrium, we tune the digestion rates to be several orders of magnitude slower than the longest polymer relaxation times of the fluids as well as the video

acquisition time. Specifically, the fastest time at which full digestion occurs is $\tau_c \approx 30$ mins compared to polymer relaxation times of $T_D \approx 2$ s and data acquisition times of 120 s. As such, the number of cleavages within one relaxation time is $\chi \equiv T_D/\tau_c \lesssim 10^{-3}$ and is $\lesssim 0.05$ within one acquisition time. Thus, we can consider our system to be in quasi-steady state with respect to the architectural degrees of freedom during each of our time-resolved measurements, allowing for unambiguous determination of diffusion coefficients and viscosity.

Further, in experiments that examine the rheology of blends with different fractions of linear and ring DNA, the two solutions comprising the distinct topologies are mixed at different ratios. Each data point is from a different sample in a different chamber and the various mixtures are ‘man-made’. This process inevitably introduces error through concentration variations and mixing inconsistencies. Further, the number and precision of the different topological ratios is rather limited due to small sample quantities (the extraction and purification process is both time-intensive and resource intensive) and extended data acquisition times. Here, the different ratios of the topologies are achieved via in situ RE digestion on the exact same sample without human intervention allowing for the **~450 measurements** we show in this paper. Finally, ‘mixing’ of the different topologies occurs throughout the sample at the molecular level via polymer relaxation, facilitating homogeneous mixing.

The reviewer is correct that we have previously published experimental and theoretical works examining the rheological and structural properties of ring-linear DNA blends and ring-supercoiled DNA blends. However, we have not investigated rheological implications of converting supercoiled to linear conformations (or blending these two topologies), or the role of reducing polymer length and/or introducing length polydispersity on the rheology. *We now emphasize these points in the text.*

3) Although the authors have presented rheological data over time, the temporal evolution of the properties is not linked appropriately with the corresponding topological changes. In this regard, the simulations data is sparse, and could have provided valuable insights. The section on the MD simulations reads considerably truncated, as only a very small part of the experimental data is captured through the simulation results. The angle of programmable rheology could have been extensively captured through the simulations, though the authors have not provided any insight therein. Finally, the simulations could come handy in establishing the topology-rheology link, an angle which has been left out despite its central premise in this work.

We thank the reviewer for the important suggestion to provide more direct connection between topology and rheology and expand the simulation results to aid in this effort. As we describe above, we now directly link topology to rheology in **Figs 1-4** and throughout the text. We have also considerably expanded our section describing the simulation results and have included new analyses that specifically investigate the changing conformational size of the polymers as a result of cutting a certain fraction of the supercoiled chains.

As we describe in detail above, we now explicitly show the dependence of rheology on the topological composition of the fluids at each time point (which we quantify by the fraction of linearized chains), and provide discussion of the results. To determine the physics underlying the relationship between rheology and topology we use our MD simulations to determine the dynamics of overlapping supercoiled polymers resulting from linearizing a fraction ϕ_L . Our results, shown in **Figs 2 and 3** indeed corroborate our experimental results, showing a slowing of MSDs and diffusion coefficients with increasing ϕ_L , an effect that is amplified at higher DNA concentrations, as we see in experiments. Motivated by the reviewer’s comments, we performed new analyses on our simulation data to measure

the average radius of gyration $\langle R_g \rangle$ and the root-mean square end-to-end distance $\langle R_{ee} \rangle$ of the polymers over the simulation time. As shown in **Fig 2**, we find that both $\langle R_g \rangle$ and $\langle R_{ee} \rangle$ increase with increasing ϕ_L , and the functional dependence aligns with our experiments, as shown in **Fig 3**.

We describe these results in the main text as follows:

As shown in Fig 2, we find that, in agreement with our microrheology data (Fig 1), the larger the fraction of linearized DNA, the monotonically slower the dynamics, which we quantify by computing the diffusion coefficient of the centre of mass of the chains at large times $D(\phi_L)$, normalized by its value when there are no cut chains $D_0 = D(\phi_L = 0)$. As shown in Fig 2d, the reduction in D/D_0 is stronger for the higher DNA concentration, as with the rheopecty in our experiments, with the effect being most pronounced at higher ϕ_L (corresponding to longer digestion times).

To shed light on the topological effects that give rise to this slowing down, we measure the radius of gyration R_g averaged over all simulated chains, as a function of simulation time t_s (Fig 2). As shown, immediately after cutting, there is a drop in $\langle R_g \rangle$ as the supercoiled, linear-like chains begin to unravel and segments adopt more entropically-favorable configurations (Fig 2e). As the chains continue to unravel they once again swell as they assume random coil configuration (Fig 2f). This non-monotonic conformational uncoiling is most apparent at $\phi_L = 0.9$ and $\phi_L = 1$, where the polymer dynamics are slow and complete equilibration is not reached until the last ~30% of the simulation time (signified by the time-independent plateau in $\langle R_g \rangle$). We also compute a complementary metric of the conformations assumed by the chains, the average root-mean-square end-to-end distance $\langle R_{ee} \rangle$, which shows similar trends as $\langle R_g \rangle$.

To directly correlate dynamics with polymer conformations in our simulations we plot D/D_0 as a function of $\langle R_{ee} \rangle$ (Fig 2i), which clearly shows that the increase in coil size dictates the slowing down of dynamics. Importantly, the degree to which diffusion is slowed with increasing $\langle R_{ee} \rangle$ is stronger for higher DNA concentrations, indicating the contribution of polymer entanglements in the rheopectic behavior, similar to experiments (Fig 1).

Importantly, we now include an entirely new figure (Fig 3) that shows new analyses that directly compare simulation and experiment results. We compute experimental diffusion coefficients from our microrheology data and experimental $\langle R_g \rangle$ values from our gel electrophoresis analysis and known topology-dependent values of R_g and compare to simulation results as a function of ϕ_L . Our simulated and experimental data generally follow similar dependences on both ϕ_L as well as DNA concentration. *We describe these new findings in the text as follows:*

To directly compare our simulations and experiments, we evaluate the experimentally measured diffusion coefficients for our 1 μm beads as a function of ϕ_L and compare the corresponding D/D_0 values with the results from simulations shown in Fig 2. As shown in Fig 3a, both experimental and simulated diffusion coefficients drop by ~2-5-fold as ϕ_L increases to 1, and this decrease is greater at higher DNA concentrations. Simulations do show a slightly weaker dependence of D/D_0 on ϕ_L and concentration, as compared to our experimental data, which we attribute to the lack of open rings, and hence reduced threadings, in simulations. Recall that our simulated polymers start fully supercoiled and are cut to linear chains, whereas our experimental solutions have open rings, which appear to be cut at a slower rate than the supercoiled constructs (see Fig 1e-g). As such, we expect that the less pronounced slowing down in the simulations as compared to experiments may be due to fewer threading events which slow dynamics of entangled rings appreciably more than reptative dynamics alone.

As our simulations indicate that the changing conformational size of the polymers, as well as the degree of entanglement (i.e., c), are driving factors in the rheology we measure, we next compute $\langle R_g \rangle$ as a function of time for the data shown in Fig 1. We use previously reported R_g values and relations $R_{g,L}/R_{g,R} \approx 1.6$ and $R_{g,L}/R_{g,SC} \approx 2.1$, along with ϕ_L values quantified from gel electrophoresis analysis to compute $\langle R_g \rangle$, averaged over all topologies, as a function of digestion time t_d and linearised fraction ϕ_L . As shown in SI Fig S5, $\langle R_g \rangle$ increases with ϕ_L , similar to simulations. The impact of increasing $\langle R_g \rangle$ on the dynamics in experiments and simulations is captured in Fig 3b, in which we plot D/D_0 as a function of $\langle R_g \rangle$. As shown, D/D_0 decreases with increasing $\langle R_g \rangle$, with remarkably similar function dependence for experiments and simulations. Notably, in both cases, the more rapid drop in D/D_0 for higher DNA concentrations is stark. Moreover, in all cases, the dependence is stronger than the $D \sim R^{-1}$ Stokes dependence expected in the dilute case for both simple spheres as well as ring, supercoiled and linear polymers [..], once again pointing to the impact of polymer overlap in the rheology.

4) Since the ER is expected to progressively alter the structure of the DNA complex fluid, how do the authors rationalize the rheology data with the partial transformation of the structures (at any intermediate time point in the ER)? For instance, can the authors fit the rheology data with the “effective structure” of the DNA fluid at any instant of experiment? Importantly, what would be the corresponding topological metric of the system? How do the authors link DNA topology (which is a local property at any instance of ER) and the observed rheology (which, for the relative scales, may be a bulk property) at any time point of the ER.

We thank the reviewer for the suggestions. We first point out that the DNA comprising our fluids are flexible chains that adopt random coil configurations at the concentrations and buffer conditions used here. The coils are isotropically distributed throughout the solution so there is no overall ‘structure’ to the fluid, in contrast to nematic fluids and liquid crystals. In our fluids, effective structure metrics are the topological makeup (i.e. linear fraction ϕ_L), the degree of polymer overlap (c/c^*) and the average size of the DNA coils (i.e., $\langle R_g \rangle$).

The plots in **Figs.1 and 4**, in which we show that all our viscosity data collapse onto master curves as a function of c/c^* or against rescaled time, clearly indicate that the relevant metrics of our fluids are c/c^* and the average fragment length $\langle l_f \rangle$ at any point during the digestion.

We now explicitly correlate our measured viscosity values to these metrics in **Figs 1-4**. We link local DNA topology to bulk rheology by considering the fraction of linearized chains at each time point (a bulk property determined from gel electrophoresis) as well as the corresponding degree of overlap (c/c^*) which is derived from the time-varying topology dependent $\langle R_g \rangle$ (a local property).

Consequently, have the authors observed spatial variations in their microrheology experiments (at any intermediate time point), or do the authors observe spatially homogeneous rheology?

We thank the reviewer for asking this question. The spatial variations that we observe are captured in the error bars shown. Specifically, we quantify error by computing MSDs and resulting diffusion coefficients and viscosities for 5 random subsets of each dataset and computing the standard error across the subsets. We have added this information to the Methods section. The spread in the data is substantially larger than controls without REs (added now in SI **Fig. S2**), suggesting that, while

digestion is slow on the timescale of molecular relaxation, the mixing timescale may be on the order of the data acquisition time. The noisiness in the data may also be an indication of transient threading of circular polymers as we describe in the manuscript as follows:

Importantly, while the increase in viscosity is generally monotonic over the course of the digestion time, the trend is 'noisy' with dips and valleys for all but the lowest DNA concentration. Notable features are the noise in the plateau of the viscosity for the (3 mg/ml, 0.5 U/ug) case that includes a small decrease over the last ~40 minutes. The noise in the time dependence is coupled with larger error in the data at each time point, suggesting heterogeneous transport modes. Such heterogeneity, which is significantly amplified compared to the control data without REs (see SI Fig S2), has been previously predicted and observed in entangled rings and ring-linear blends [..] and attributed to ring-ring and ring-linear threading events as well as modified reptation of folded and branched ring conformations [..].

Other potential indicators of threading are the steep upticks in the viscosity for the (6 mg/ml, 0.05 U/ug), (3 mg/ml, 0.05 U/ug) and (3 mg/ml, 0.10 U/ug) cases which occur at times that correspond to ~15% and ~70% linear chains. The low linear fraction is comparable to the fraction of linear chains needed in ring polymers to markedly reduce ring diffusion via threading [..], while the high fraction is similar to that previously reported as necessary for a large increase in the plateau modulus and complex viscosity that peaks at ~70% linear chains before decreasing again [..].

Further, our gel electrophoresis analysis shows that the supercoiled DNA is digested at a faster rate than the ring DNA (Fig 1e-g), so the effect of ring-linear threading would be more significant at later times in the digestion where, indeed, the features described above are most apparent. Finally, we note that while threading of open rings has been well-documented in the literature, it has been suggested that tightly wound supercoils are much less likely to be threaded by neighboring chains [..] owing to their closed conformations. However, threading of rings by supercoils may play a role in increased viscoelasticity in ring-supercoiled blends [..]; and the highly branched, amoeba-like structures that supercoiled polymers adopt may also lead to heterogeneous transport modes.

5) Structural changes due to ERs are expected to occur in 3D, although the authors quantify particle mobility (in the microrheology experiments) using 1D mean squared displacement. The authors should justify this difference in dimensionality, and comment on its impact in the interpretation of the presented data.

We thank the reviewer for requesting clarity on this point. We actually track the microspheres in 2D and average over the two planar dimensions, after ensuring they are similar, to get the 1D MSD we present. We now make this point clearer in the text. Further, we only track beads that are in focus, so their movement in z (while tracking in x and y) is limited to roughly their radius (500 nm) which correlates to an MSD of order $10^{-1} \mu\text{m}^2$, which is below the large lag time MSD data that we use to determine diffusion coefficients and viscosity values. Nonetheless, because we are not considering motion in z, it is possible that MSDs are systematically slightly underestimated, which we now point out in the Methods section. However, this effect should have minimal impact on the metrics we extract from the MSDs or the trends we present. Finally, because the beads we track have a 500 nm radius, ~2-10x larger than the radius of gyration of any of the DNA molecules or fragments examined in this work, any conformational changes to the DNA that occur in z should be captured in the 2D trace of the bead. *We have added this information to the Methods section.*

6) The title and motivation of the manuscript limits the scope of the manuscript. The authors should bring out the significance of their results clearly (possibly along the above suggested points), which together with additional experiment and simulations, could enhance the novelty of the presented results.

We thank the reviewer for the suggestion. We have changed the title and introduction section to better motivate our work and highlight its novelty. Specifically, we have:

1. changed the title to 'Topological digestion drives programmable rheopecty and thixotropy in entangled DNA fluids';
2. amplified the Introduction to include discussion of circular and linear polymers in general, beyond DNA, focusing on significant open questions that our work addresses and potential applications;
3. changed the abstract to reflect the more generalized and encompassing introduction;
4. highlighted the novelty of investigating supercoiled polymers and topological blends;
5. clearly delineated the differences between our system and other efforts in out-of-equilibrium materials, in particular those that incorporate topology, such as topological microfluidics.
6. clearly stated that without these time-resolved experiments, it would have never been possible to quantitatively determine the rate of change of rheology, or its coupling to the composition on the resolution scale that we achieve here.

Minor comments:

- 1) The authors should spell out RE at its first appearance in the text
- 2) k in characteristic cutting time should be defined at its first point of appearance
- 3) Figure numbers are styled/formatted inconsistently, the authors could make these consistent

We thank the reviewer for drawing our attention to these oversights. We have made the suggested edits.

Reviewer #2 (Remarks to the Author):

The paper studies how topological and architectural alterations of DNA solutions driven by endonuclease reactions control the rheology of the fluids. The key findings of the paper include: 1. entangled linear DNA solutions undergoing restriction enzyme digestion display a marked decrease in viscosity; 2. circular DNA solutions undergoing restriction enzyme digestion display a monotonic increase in viscosity; 3. combining restriction enzyme of different cutting frequencies, the viscosity of circular DNA solutions first increases due to entanglement of linearized chains and then decreases due to further fragmentation of the linear chains. The experiments were explained by scaling analysis and supported by molecular dynamics simulations.

The results are solid and the experiments/analysis were done in an expert manner. I found the paper very interesting while reading the abstract/introduction and was looking forward to see results on "a wide range of non-equilibrium rheological behaviours that can be programmed". But in the end, the authors only briefly touched on the programmable dynamics (Fig. 5). The general principle of using enzymatic reactions to control molecular topology and flow properties of DNA solutions dates back to 2010 (PMCID: PMC2941032), and recent publications from the senior author's lab have reported very interesting findings along this line. Although restriction enzymes are now used instead of topoisomerase, the general principle is similar. Also restriction enzyme digestion is known by molecular biologists to cause reduction of viscosity; for example, see this article PMCID: PMC4948299 found by a quick search. Results in Fig. 5 are straightforward. A more creative and systematic design of programmable dynamics of controlled fluid rheology (including viscoelasticity, another important aspect overlooked by the authors) would substantially strengthen the paper. I really like the aspirational discussion in the paper, but results presented in the current paper seems not yet ready to serve the aim.

We thank the reviewer for finding our work 'solid' and 'done in an expert manner' and 'very interesting'. We address the other comments below.

We realize that in the original manuscript we did not make clear the programmability of our results nor their 'wide range...that can be programmed' by the concentrations, sizes, and initial topologies of the DNA as well as the concentrations and types of REs. The programmability of our materials is reflected in all of our experiments, not just Fig 5, as the degree, rate and time-course of rheopecty and thixotropy depend on the DNA and RE concentrations as well as the polymer overlap and conformational size. These are all tuning knobs that can be used to program different out-of-equilibrium rheological properties. However, in the original manuscript we chose to focus on explaining the underlying physics and establishing universal scaling relationships - essential to being able to predict and program diverse rheological properties of DNA-RE fluids by varying the molecular input parameters. *We have now substantially rewritten much of the text to make these points clearer.*

The reviewer aptly points to several works that show the use of enzymatic reactions to control flow properties, which we include in our Introduction. However, we note that PMCID: PMC2941032 uses topoisomerases to transiently remove entanglements in entangled **linear** DNA solutions. The topoisomerase, fueled by ATP, cuts the linear strands, unwinds them then reconnects them. The topology and size of the **linear** DNA are unchanged and the final state of the fluid, after the polymers relax, is the same as the starting state. This is quite a different system than ours in which we permanently alter the topology and/or length of the polymers to drive a transition to a new steady-state. *We now discuss this work in the manuscript as follows:*

We note that previous experiments have shown that Topoisomerase II can reduce the viscoelasticity of entangled, similarly to what we show here. In these experiments, Topoisomerases relied on ATP to disentangle linear lambda DNA, by cleaving and reconnecting individual strands without irreversibly changing its structure, size or topology. Further, this effect was transient and the system eventually returned to its initial steady-state. In contrast, here we drive the system to an entirely new steady-state by permanently altering the size and degree of entanglement of the comprising DNA molecules. It would be interesting in the future to add ligase enzymes to our materials to trigger the fusion and repair of the DNA fragments.

Our previous works that the reviewer refers to have focused on the rheology of steady-state mixtures of (i) concentrated 45 kbp ring and linear DNA and (ii) semidilute 25 kbp ring and supercoiled DNA. In (i) we examine the effect of linear fraction on the rheology, similar to our analysis here; however, we only examine a single length and concentration of DNA and there is no supercoiled DNA present. Note that in this previous study, the steady-state nature of the experiments made them resource and time intensive and more prone to uncertainty (each data point is from a different sample, mixed separately) limiting the number of formulations (defined by ϕ_L and c/c^*) we could measure (6 formulations vs **240** measured in Figs 1 and 5 of this paper). We also note that our results here are different than those reported in (i) in which we found a non-monotonic dependence of the rheological properties on ϕ_L , rather than a monotonic dependence that we measure here, which we attributed to threading of rings. Beyond the novelty of the in situ topological conversion in our current work, we also examine smaller supercoiled DNA in more dense states in which we expect the effect of threading to be quite different, as *we now discuss in the main text (e.g., in the excerpt below)*. Such a system had yet to be studied even in steady-state, and the role of supercoiling in polymer rheology remains scarcely studied and poorly understood.

Importantly, while the increase in viscosity is generally monotonic over the course of the digestion time, the trend is 'noisy' with dips and valleys for all but the lowest DNA concentration. Notable features are the noise in the plateau of the viscosity for the (3 mg/ml, 0.5 U/ug) case that includes a small decrease over the last ~40 minutes. The noise in the time dependence is coupled with larger error in the data at each time point, suggesting heterogeneous transport modes. Such heterogeneity, which is significantly amplified compared to the control data without REs (see SI Fig S2), has been previously predicted and observed in entangled rings and ring-linear blends [...] and attributed to ring-ring and ring-linear threading events as well as modified reptation of folded and branched ring conformations [...].

Other potential indicators of threading are the steep upticks in the viscosity for the (6 mg/ml, 0.05 U/ug), (3 mg/ml, 0.05 U/ug) and (3 mg/ml, 0.10 U/ug) cases which occur at times that correspond to ~15% and ~70% linear chains. The low linear fraction is comparable to the fraction of linear chains needed in ring polymers to markedly reduce ring diffusion via threading [...], while the high fraction is similar to that previously reported as necessary for a large increase in the plateau modulus and complex viscosity that peaks at ~70% linear chains before decreasing again [...].

Further, our gel electrophoresis analysis shows that the supercoiled DNA is digested at a faster rate than the ring DNA (Fig 1e-g), so the effect of ring-linear threading would be more significant at later times in the digestion where, indeed, the features described above are most apparent. Finally, we note that while threading of open rings has been well-documented in the literature, it has been suggested that tightly wound supercoils are much less likely to be threaded by neighboring chains [...] owing to their closed conformations. However, threading of rings by supercoils may play a role in increased

viscoelasticity in ring-supercoiled blends [..]; and the highly branched, amoeba-like structures that supercoiled polymers adopt may also lead to heterogeneous transport modes.

In (ii), in an effort to shed light on the effect of supercoiling on rheology, we measured the rheological properties of semidilute blends of ring and supercoiled 25 kbp DNA. We showed that these blends exhibited surprising stiffness and elastic properties despite their low entanglement density, which we attributed to threading of rings by supercoiled chains and improved flow alignment of rings. However, in this study, we did not vary the fraction of rings and supercoiled constructs and the concentrations were all $\sim c^*$ - a very different regime than that probed here, with results that differ from what we report here. *We include the following text in the manuscript to address this point:*

Far less understood are the rheological properties of supercoiled polymers, for which DNA is an archetypal example [..]. Previous microrheology studies on semidilute blends of ring and supercoiled DNA have shown that blends exhibit entanglement dynamics at concentrations well below that needed for monodisperse systems of ring or linear polymers to exhibit similar viscoelastic properties [..]. At the same time, simulations of dense supercoiled and ring polymers have shown that supercoiled DNA molecules exhibit faster diffusion and more swollen conformations than their ring counterparts [..].

Finally, the reviewer is correct that biologists have long appreciated that digestion can **decrease** the viscosity of DNA fluids; however, no works have reported an increase in viscosity (**Figs 1-3**) and/or gated reduction (**Fig 5**). *We have added the following statement to the text to clarify this point:*

Importantly and intriguingly, we highlight that the observed rheopectic behavior during digestion is in marked contrast with the fluidization of DNA solutions typically observed during digestion [..]. Perhaps the first quantitative record of this phenomenon was reported in 1970, when Welcox and Smith used an Ostwald viscometer to measure the change in viscosity of a solution of viral P22 DNA mixed with Haemophilus Influenzae lysate [..]. They measured that the solution became less viscous over time, strongly suggesting the existence of a 'restriction' enzyme within the bacterium lysate that was cutting the P22 DNA. This was then identified as HindIII, the first restriction enzyme ever discovered and for which Smith was awarded a Nobel Prize in Medicine [..]. Since then, DNA digestion has been commonly associated with a decrease in solution viscosity [..]. It is thus quite notable that our solutions of circular DNA cut at only one site present such marked increase in viscosity.

Moreover, the time course of the out-of-equilibrium transitioning from the initial to final state, and its dependence on RE and DNA concentrations and sizes had yet to be discovered or even considered, despite its importance in digestion reactions. *We have now made this point clearer throughout the text.*

Below are some comments for the author's consideration:

1. The authors tried to make connection to active matter. There is no local free energy input in the experimental system studied in this paper, and although the system can just be called as out-of-equilibrium, the dynamics is driven by free energy minimization like in all regular chemical reactions. The connection to active matter sounds vague, unnecessary, and even confusing, especially in view of the many existing studies on self-propelled polymers.

We thank the reviewer for their candid perspective on describing our system. We agree with the reviewer that there is no local free energy input, *which we now emphasize in the main text*. We also limit reference to active matter and clearly delineate the differences between our system and other out-of-equilibrium materials. Our nomenclature of 'topologically-active' is meant to differentiate our

platform from these other systems that rely on energy input to drive translational motion, such as self-propelled polymers, as suggested by the reviewer. The topologies of our molecules are changing over the course of the experiment, and it is this change that drives the time-varying rheology. We have reworded the Introduction and Conclusions to more carefully classify our system and compare to existing materials. *Some sample excerpts include:*

'Non-equilibrium systems that undergo self-driven structural or rheological changes are of intense current interest as a platform for designing next-generation multifunctional materials [..]. Typically, systems are pushed out of equilibrium by dissipating energy to drive the movement of their constituents [..]. On the contrary, using topological alterations to drive systems out of equilibrium remains largely unexplored [..]. Such "topologically-active" systems harness topological or architectural changes in their macromolecular constituents to drive time-varying rheological properties of the bulk material.'

'Our approach of harnessing topological conversion to drive rheological transitions complements recent efforts that explore the connection between topology and non-equilibrium dynamics, such as self-propelled rings [..], active nematics with topological defects[..], topological microfluidics [..], and active Olympic gels [..]. Notably, the "topologically-active" fluids we study here are pushed out-of-equilibrium by changes in the topology and architecture of the constituents, in contrast to previous studies that couple translational activity to topologically non-trivial systems, such as ring polymers [..], bacterial suspensions [..], or liquid crystals [..]. To establish the generic framework of "topologically-active" DNA-based materials, here we focus on the design and characterization of "one-shot" systems that start and stop in steady-states. Our future works will build on this framework and incorporate energy-consuming topological processes and reversibility into the fluids, as well as coupling the fluids to other synthetic and biological systems to create non-equilibrium matter with exotic rheological properties. This platform may be leveraged for diverse materials and biomedical applications from drug delivery, filtration and sequestration to self-curing and infrastructure repair.'

2. The authors could elaborate how viscoelasticity of the fluids were affected or controlled.

We thank the reviewer for raising this question. We indeed examined the viscoelasticity of the fluids by using the generalized Stokes Einstein relation (GSER) framework to convert our measured MSDs to linear viscoelastic moduli, $G'(\omega)$ and $G''(\omega)$ and the corresponding complex viscosity $\eta^*(\omega)$. However, most of our data exhibited very little elasticity and instead exhibited nearly Newtonian viscous fluid properties, in particular at lower frequencies. As such, we focused our analysis on the zero-shear viscosity, as derived from fits to the MSDs at large lag times where they scale linearly with time. *We now make this point clearer in text.* Nevertheless, while there is minimal viscoelasticity over most of the measurement window, we do observe modest elasticity and shear thinning at later digestion times and the highest DNA concentration for the linearizing circular DNA (**Fig 1**) and early times for the fragmenting entangled linear DNA (**Fig 4**). *We now show these data in Figs 1 and 4, and discuss the results in the text as follows:*

'We note that as t_a increases, some of the MSDs transition from exhibiting purely free diffusion, namely $\text{MSD} \sim t^\alpha$ with $\alpha=1$, to displaying modest subdiffusion (i.e., $\alpha \simeq 0.9$) at short lag times (~ 0.1 seconds), suggestive of the onset of high-frequency viscoelasticity.

To better characterize potential viscoelastic behavior, we compute the frequency-dependent complex viscosity $\eta^*(\omega)$ computed from the MSDs using the generalised Stokes-Einstein relation (GSER) [..] (detailed in the Methods). While most of the solutions exhibit largely Newtonian behaviour during digestion, manifesting as frequency-independent $\eta^*(\omega)$, the 6 mg/ml solution exhibits high-frequency viscoelasticity for $t_a > 15$ mins, reflected by the shear-thinning behavior (i.e. a decrease of $\eta^*(\omega)$ with ω) that increases with increasing t_a (Fig. 1m). In the cases in which we observe high-frequency viscoelasticity, we restrict our analysis to frequency and lag-time values in which $\eta^*(\omega)$ is ω -independent and MSDs scale linearly with lag time.'

'Analogous to what we see in Fig. 1, some MSDs display a subdiffusive regime at short lag times t , reflecting the viscoelasticity of entangled λ DNA. After $t \approx 2$ s, comparable to the longest relaxation time (i.e., the reptation or disengagement time of our λ DNA solutions [..], MSDs crossover to free diffusion. On the contrary, fully digested fluids display free diffusion across all lag times, suggesting minimal viscoelasticity Fig 4c. Similar to Fig. 1m, we characterize the viscoelasticity as a function of digestion time by computing the complex viscosity $\eta^*(\omega)$ from the MSDs using GSER. Fig. 4d indeed shows high- ω viscoelasticity (evidenced by shear-thinning) that becomes progressively weaker as t_a increases.'

'From D , we compute the zero-shear viscosity η using the Stokes-Einstein equation...Strictly speaking, this approach is only valid for Newtonian fluids that exhibit purely viscous behavior, and the highest DNA concentrations we examine exhibit modest viscoelasticity and subdiffusion at short lagtimes for certain digestion times t_a (Figs. 1m, 4d). However, for these cases we restrict our analysis to the large-time terminal regime in which the MSD scales linearly with lag-time and the complex viscosity is independent of frequency.'

3. Certain points discussed in Conclusion are a bit overstretched given the existing results.

We thank the reviewer for the suggestion. We have edited the Conclusions accordingly, as follows:

Specifically, we show that it is the time-varying degree of polymer overlap at any point in time, which depends on the changing topologies of the DNA molecules, that programs the rheopecty of circular DNA undergoing linearization. Conversely, we observe thixotropy of entangled linear DNA undergoing fragmentation that is tuned by the time-varying average fragment length which is dictated largely by the RE concentration. Finally, we demonstrate the facile coupling of these two behaviors - rheopecty and thixotropy - by engineering DNA fluids to exhibit non-monotonic gated rheology, such as initial rheopecty followed by time-gated thixotropy which can be programmed by judicious tuning of the RE cocktail.

The versatile and tunable self-driven systems that we present can be harnessed for diverse applications from self-healing tissue to controlled drug delivery. For instance, time-controlled targeted drug release may be engineered utilising the gated thixotropy we demonstrate (Fig. 5; and autonomous tissue healing can be made possible by rheopecty of biomimetic complex fluids (Fig. 1) at a wound site. Further, in order to design the next generation of drug delivery systems or degradable scaffolds for tissue regeneration, it is necessary to understand and develop predictions and relations, as we do here, regarding how the viscosity of DNA materials change during enzymatic digestion.

Our work may also shed new light on the viscoelasticity of entangled genomic material in key cellular processes such as replication and gene expression. Finally, and perhaps most importantly, our studies

showcase the potential of leveraging non-trivial alterations to the structure and topology of DNA as a unique route to design and functionalise out-of-equilibrium materials.

4. The authors called the experimental system as "topologically-active fluids". In the first part of the results (cutting linear chains shorter), I do not see the change of molecular topology.

The reviewer is correct that in the fragmentation experiments, we are changing the DNA length rather than the topology. However, the premise of using REs to drive structural or architectural change to the DNA remains. We performed the fragmentation experiments to shed light on possible routes for designing gated thixotropy (**Fig 5**), and to determine how we can use time-varying length, as well as topology, to drive rheological alterations. However, because the focus of the study is on topological alterations we have reorganized the paper and figures to present the linearization experiments (circular to linear topology) and simulations first, followed by the fragmentation studies and non-equilibrium living polymer theory.

5. The authors use gel electrophoresis to determine the topology forms of DNA molecules. Could the authors provide any microscopy examination or references to help readers understand the molecular form of the bands?

We thank the reviewer for requesting further clarification and explanation of our gel electrophoresis analysis. While we typically use gel electrophoresis to determine topology, in previous works we have used single-molecule flow experiments to quantify the fractions of supercoiled, ring and linear DNA in semidilute DNA solutions. In this previous work (DOI: 10.1039/C9SM01767D), we showed that the results from the single-molecule experiments were similar to those we obtain via gel electrophoresis, and showed the different molecular conformations of supercoiled, ring and linear DNA. We now include in SI a new Fig S4 that reproduces: (i) microfluidics images from DOI: 10.1039/C9SM01767D that show the conformations of the different DNA topologies (ring, supercoiled, linear), (ii) a gel of pYES2 from the original paper in which the purification methods were first described (doi:10.1021/ma0601464) to show the mobility of the different topologies with respect to the lambda-HindIII marker, and (iii) a sample gel band analysis image from arXiv:2112.02720 showing how the relative fractions of different DNA topologies in a sample are determined.

We have also added the following text to the methods section:

'We quantify DNA concentration and topology via gel electrophoresis as described below (see Figs 1, 4, S3). We note that while the bulk nature of gel electrophoresis quantification limits its precision to within ~5 -10%, we have previously shown that using single-molecule microfluidics experiments to determine the fraction of each topology comprising our DNA solutions yields results that are comparable (within ~2-3%) to those we measure via gel electrophoresis (Fig S4).'

6. A more detailed version of scaling analysis should be provided somewhere, such as SI.

We thank the reviewer for the suggestion. We realize now that we were quite brief in our descriptions in the interest of trying to include the immense amount of data and analysis we wanted to present (**450 experimental data points, MD simulations and substantive theory development**). However, this brevity came at the cost of clarity in many instances. We now expand our discussion of the scaling

arguments in the main text. We chose to put them in the main text rather than the SI as we see their derivations as fundamental to understanding their implications. Sample text includes:

In semidilute and concentrated solutions, changing R_g directly alters the degree of overlap via the relation $c^* = (3/4\pi)(M/N_A)/(R_g^3)$ where c^* is the polymer overlap concentration... To compute c^* during digestion, in which we have a mix of different topologies – each with a different R_g value – we start with the conventional expression for c^* , derived by equating the solution volume (m/c^* where m is total mass) to the total volume the molecules comprise, i.e. the total number of molecules $N = mN_A/M$ multiplied by the volume per molecule $V_m = (4\pi/3)R_g^3$. We then consider that each component contributes separately to the total volume the molecules fill in solution: $N_L V_{m,L} + N_R V_{m,R} + N_S V_{m,S} = (4\pi/3)(\phi_L R_{g,L}^3 + \phi_R R_{g,R}^3 + \phi_S R_{g,S}^3)$. The resulting expression is then: $c^* = (3/4\pi)(M/N_A)/(\phi_L R_{g,L}^3 + \phi_R R_{g,R}^3 + \phi_S R_{g,S}^3)$.

As shown in Fig 3, our η/η_0 data all collapse onto a master curve when plotted against c/c^* , with a functional form that can be described by $\eta/\eta_0 \sim (c/c^*)^\gamma$ where $\gamma \approx 0.5$ and $\gamma \approx 1.75$ at low and high c/c^* values, respectively. The crossover in scaling takes place at $c \approx 6c^*$ which we previously showed to be at the onset of entanglement dynamics for DNA solutions. The exponents agree with those reported in single-molecule tracking studies measuring the diffusion of ring and linear DNA in semidilute ($c \approx c^*$) and entangled ($c > 6c^*$) DNA solutions, respectively. Specifically, these studies showed that $D \sim c^{-1.75}$ for 11, 25 and 45 kbp ring and linear DNA at concentrations above $6c^*$, in line with reptation model predictions. Below $6c^*$, diffusion coefficients followed $D \sim c^{-0.5}$ scaling, in accord with Rouse model predictions. These exponents, compatible with the behaviour of semidilute and entangled flexible polymers, differ from those measured via bulk rheology for semiflexible polymers and for theta solvent conditions.

As such, our experiments and simulations demonstrate that the RE-driven rheopecty we measure arises from the conversion of circular to linear polymers that increases the average conformational size of the polymers, which, in turn, increases the degree to which polymers are overlapping and entangled. The effect of topology is more dramatic in the entangled regime in which diffusive processes are more strongly dependent on concentration. As we describe above, our results shown in Fig 1 suggest that threading may contribute to the dynamics we measure, in which case we expect $D \sim c^{-3}$ rather than $D \sim c^{-1.75}$, as previously reported for ring-linear DNA blends in which constraint release from threading dominated the diffusivity. Indeed, for the highest concentrations ($c/c^* > 10$) the scaling of η/η_0 is steeper than $\gamma \approx 1.75$ and more closely follows $\gamma \approx 3$, indicating contributions from threading.'

'Eq 2 predicts that (i) the viscosity should decrease over time irrespective of RE (as seen in our experiments in Fig 4), (ii) the key digestion rate $\kappa = \chi/T_d = 1/\tau_b$ should be proportional to RE concentration (as seen in Fig 4), and (iii) the normalized viscosity should scale as $(\kappa * t_a + 1)^{-3}$. By fitting each of our experimental normalised viscosity curves in Fig 4 to $\eta/\eta_0 = (\kappa * t_a + 1)^{-3}$, we obtain a direct measure of the corresponding digestion rate $\kappa = \chi/T_d = 1/\tau_b$. Remarkably, upon rescaling the time axis by the corresponding κ value determined from each fit (i.e., $t \rightarrow x = \kappa * t_a$) all of our experimental curves collapse onto a single master curve described by $\eta/\eta_0 = (x + 1)^{-3}$. We also find that κ indeed scales linearly with RE concentration, in agreement with Michaelis-Menten kinetics and corroborating point (ii).'

7. The authors wrote "indeed, if this was the case we would expect a non-monotonic behaviour in viscosity with a maximum at low ring contaminants within a solution of linear chains [31] (expected at large digestion time) due to threading". Why a maximum at low ring contaminants? According to Ref. 29, ring threading will further increase viscosity, then why would the authors expect a non-monotonic behaviour in viscosity?

We have rewritten this section entirely and have removed this statement. However, we would expect a non-monotonic dependence on linear fraction as previous studies have shown that a small fraction of rings added to linear polymer melts increases the melt viscosity (see Parisi et al *Macromolecules* 2020). Likewise, a small fraction of linear chains added to ring polymers leads to a viscosity that is higher than that for pure linear chains of the same concentration (see Peddireddy et al *Soft Matter* 2019). These effects, resulting from threading, would lead to a non-monotonic time-dependence of viscosity in our measurements. This effect is described in the revised Introduction as follows:

While the dynamics of entangled linear polymers are well described by the reptation model developed by de Gennes, Doi and Edwards, the extension of this model to circular polymers, with no free ends, is not straightforward; and the extent to which ring polymers form entanglements, and the relaxation modes and conformations available to ring polymers remain topics of fervent debate [..].

Moreover, in blends of polymers of distinct topologies, such as ring-linear blends, the role of polymer threading and other topological interactions can lead to emergent rheological and dynamical properties such as increased viscosity, suppressed relaxation, and heterogeneous transport modes [..], compared to monodisperse systems of rings or linear chains. For example, solutions of concentrated ring polymers exhibit lower viscosity than their linear counterparts [..], while blends of ring and linear polymers exhibit higher viscosity than pure linear chains [..] and display unique behaviours under extensional stress [..].

However, the important distinction between these past studies and ours is that here we are examining relatively short chains (~60 Kuhn lengths) that start in majority supercoiled rather than ring form. Nevertheless, upon closer examination of our data and the literature, we argue that threading does in fact contribute to the rheopectic behavior we measure, despite the fact that the dependence on linear fraction is distinct from that reported in entangled ring-linear blends. *We describe the evidence of threading throughout the Results section, e.g.,:*

Importantly, while the increase in viscosity is generally monotonic over the course of the digestion time, the trend is 'noisy' with dips and valleys for all but the lowest DNA concentration. Notable features are the noise in the plateau of the viscosity for the (3 mg/ml, 0.5 U/ug) case that includes a small decrease over the last ~40 minutes. The noise in the time dependence is coupled with larger error in the data at each time point, suggesting heterogeneous transport modes. Such heterogeneity, which is significantly amplified compared to the control data without REs (see SI Fig S2), has been previously predicted and observed in entangled rings and ring-linear blends [..] and attributed to ring-ring and ring-linear threading events as well as modified reptation of folded and branched ring conformations [..].

Other potential indicators of threading are the steep upticks in the viscosity for the (6 mg/ml, 0.05 U/ug), (3 mg/ml, 0.05 U/ug) and (3 mg/ml, 0.10 U/ug) cases which occur at times that correspond to ~15% and ~70% linear chains. The low linear fraction is comparable to the fraction of linear chains needed in ring polymers to markedly reduce ring diffusion via threading [..], while the high fraction is similar to that previously reported as necessary for a large increase in the plateau modulus and complex viscosity that peaks at ~70% linear chains before decreasing again [..].

Further, our gel electrophoresis analysis shows that the supercoiled DNA is digested at a faster rate than the ring DNA (Fig 1e-g), so the effect of ring-linear threading would be more significant at later times in the digestion where, indeed, the features described above are most apparent. Finally, we note that while threading of open rings has been well-documented in the literature, it has been suggested that tightly wound supercoils are much less likely to be threaded by neighboring chains [..] owing to their closed conformations. However, threading of rings by supercoils may play a role in increased viscoelasticity in ring-supercoiled blends [..]; and the highly branched, amoeba-like structures that supercoiled polymers adopt may also lead to heterogeneous transport modes.'

'As we describe above, our results shown in Fig 1 suggest that threading may contribute to the dynamics we measure, in which case we expect $D \sim c^{-3}$ rather than $D \sim c^{-1.75}$, as previously reported for ring-linear DNA blends in which constraint release from threading dominated the diffusivity. Indeed, for the highest concentrations ($c/c^* > 10$) the scaling of η/η_0 is steeper than $\gamma \approx 1.75$ and more closely follows $\gamma \approx 3$, indicating contributions from threading.'

'The time-dependence in the thixotropic phase [referring to green data in Fig 5b] is noisy compared to the fluids with higher EcoRI concentrations, and the error at each time point is likewise higher. This phenomenon, similar to what we see in Fig. 1, is suggestive of threading of rings, as we describe above.'

8. The authors described XhoI as 2-cutter and EcoRI as 10-cutter because the RE cut twice or 10 times on the plasmid. This is really confusing, because the numbers in n-cutter refers to the length of recognition site of an RE.

We thank the reviewer for noting this point of confusion. We have removed this nomenclature from the paper.

9. Fig. S3: Please show the full ladder markers to help readers better tell the mobility of different bands. Fig. S4: Please indicate what each band is.

We have made the requested edits.

10. Fig. S5: Ref 3 did not show data for 0.5 g/ml DNA concentration.

We thank the reviewer for pointing out this imprecision. We were reproducing the data from Ref 3 for 0.4 mg/ml and were comparing it to our measured viscoelastic moduli at 0.45 mg/ml to benchmark our GSER analysis. However, since it was not referenced in the main text, we have now removed it from the SI.

Reviewer #3 (Remarks to the Author):

Dear Editor,

This manuscript presents an elegant and timely series of experiments and simulations that explore topologically complex blends of ring and linear DNA as a rheologically programmable active fluid system. The authors use natural enzymes to actively transform DNA dispersity and topology and demonstrate that this can be engineered to significantly increase or decrease fluid viscosity in a controlled way. The authors present simple and compelling arguments to explain their linear chain digestion data, and they provide complementary simulations for the ring-digestion case where no simple arguments are at hand.

We thank the reviewer for finding our work ‘elegant and timely’ and finding our arguments ‘simple and compelling’.

While I defer analysis of the experimental details to those more qualified, I have examined the simulation methods and find no issue with them that would invalidate the authors’ core conclusions. However, I do believe that their discussion of their simulation methodology requires some additional supplemental details – as I discuss below.

We thank the reviewer for the suggestions which we address below.

In addition, the authors have kept their discussions very focused on DNA systems, but ring and ring/linear blend rheology’s have also been a popular research topic in non-DNA systems – the Journal of Rheology just released a special issue with many studies in this area. To emphasize the broad impact of their study, I’d encourage the authors to add some additional context and contrast between their observations and those seen in non-supercoiled polymer blends.

We thank the reviewer for this suggestion which we believe has strengthened our manuscript. We now include a substantial section in the Introduction on synthetic ring melts and ring-linear blends. We also discuss our results in the context of some of these previous findings. In particular, we highlight the role of threading in previous non-supercoiled blends and the extent to which we think threading may play a role in our findings. Some text we have added includes:

In the materials and engineering communities, polymer topology has long been appreciated for its ability to confer novel rheological properties to polymeric fluids and blends that can be tuned for commercial and industrial use [..]. In particular, end-closure of linear polymers (creating open ring, knotted and linked constructs) and breakage and fragmentation of circular polymers (resulting in linear chains) and their roles in the rheology and dynamics of entangled polymer systems is a vibrant and widely-investigated topic of research [..]. While the dynamics of entangled linear polymers are well described by the reptation model developed by de Gennes, Doi and Edwards [..], the extension of this model to circular polymers, with no free ends, is not straightforward. The extent to which ring polymers form entanglements, and the relaxation modes and conformations available to ring polymers remain topics of fervent debate [..].

Moreover, in blends of polymers of distinct topologies, such as ring-linear blends, the role of polymer threading and other topological interactions can lead to emergent rheological and dynamical properties such as increased viscosity, suppressed relaxation, and heterogeneous transport modes [..], compared to monodisperse systems of rings or linear chains. For example, solutions of concentrated ring polymers exhibit lower viscosity than their linear counterparts [..], while blends of ring and linear

polymers exhibit higher viscosity than pure linear chains [..] and display unique behaviours under extensional stress [..].

Far less understood are the rheological properties of supercoiled polymers, for which DNA is an archetypal example [..]. Previous microrheology studies on semidilute blends of ring and supercoiled DNA have shown that blends exhibit entanglement dynamics at concentrations well below that needed for monodisperse systems of ring or linear polymers to exhibit similar viscoelastic properties [..]. At the same time, simulations of dense supercoiled and ring polymers have shown that supercoiled DNA molecules exhibit faster diffusion and more swollen conformations than their ring counterparts [..].

The rich and surprising behavior of topologically-distinct polymeric fluids, along with the principal role that topology plays in dictating the conformational size and structure of the comprising polymers, as well as the nature of the interactions between them, inspired us to exploit polymer topology as a route to functionalize polymeric materials with spatiotemporally varying rheological properties. We use entangled DNA as our seminal 'topologically-active' polymeric material as DNA has been extensively employed as a model system to study polymer physics [..]. Further, DNA is particularly well-suited to study the role of topology on the rheology and dynamics of polymeric fluids as it naturally occurs in supercoiled, ring and linear conformations [..]. Finally, conversions from one DNA topology to another, via enzymatic reactions, play critical roles in myriad cellular processes such as replication and repair [..].'

Importantly, while the increase in viscosity is generally monotonic over the course of the digestion time, the trend is 'noisy' with dips and valleys for all but the lowest DNA concentration. Notable features are the noise in the plateau of the viscosity for the (3 mg/ml, 0.5 U/ug) case that includes a small decrease over the last ~40 minutes. The noise in the time dependence is coupled with larger error in the data at each time point, suggesting heterogeneous transport modes. Such heterogeneity, which is significantly amplified compared to the control data without REs (see SI Fig S2), has been previously predicted and observed in entangled rings and ring-linear blends [..] and attributed to ring-ring and ring-linear threading events as well as modified reptation of folded and branched ring conformations.

Other potential indicators of threading are the steep upticks in the viscosity for the (6 mg/ml, 0.05 U/ug), (3 mg/ml, 0.05 U/ug) and (3 mg/ml, 0.10 U/ug) cases which occur at times that correspond to ~15% and ~70% linear chains. The low linear fraction is comparable to the fraction of linear chains needed in ring polymers to markedly reduce ring diffusion via threading [..], while the high fraction is similar to that previously reported as necessary for a large increase in the plateau modulus and complex viscosity that peaks at ~70% linear chains before decreasing again [..].

Further, our gel electrophoresis analysis shows that the supercoiled DNA is digested at a faster rate than the ring DNA (Fig 1e-g), so the effect of ring-linear threading would be more significant at later times in the digestion where, indeed, the features described above are most apparent. Finally, we note that while threading of open rings has been well-documented in the literature, it has been suggested that tightly wound supercoils are much less likely to be threaded by neighboring chains [..] owing to their closed conformations. However, threading of rings by supercoils may play a role in increased viscoelasticity in ring-supercoiled blends [..]; and the highly branched, amoeba-like structures that supercoiled polymers adopt may also lead to heterogeneous transport modes.'

'As we describe above, our results shown in Fig 1 suggest that threading may contribute to the dynamics we measure, in which case we expect $D \sim c^{-3}$ rather than $D \sim c^{-1.75}$, as previously reported for ring-linear DNA blends in which constraint release from threading dominated the diffusivity (REF).

Indeed, for the highest concentrations ($c/c^* > 10$) the scaling of η/η_0 is steeper than $\gamma \approx 1.75$ and more closely follows $\gamma \approx 3$, indicating contributions from threading.'

'The time-dependence in the thixotropic phase [referring to green data in Fig 5b] is noisy compared to the fluids with higher EcoRI concentrations, and the error at each time point is likewise higher. This phenomenon, similar to what we see in Fig. 1, is suggestive of threading of rings, as we describe above.'

I recommend this study for publication in Nature Communications provided the authors respond to these comments as well as those below.

We are pleased that the referee recommends publication in Nature Communications. Below we address all of their comments.

###Comments Regarding the Main Text

#Terminology "rheopecty" – In the abstract the term rheopecty is used. In the conclusion, the term anti-thixotropy is used instead. I'd recommend unifying the language used for this increasing low-rate viscosity.

We thank the referee for pointing out this point of confusion. Throughout the text we now refer to increasing and decreasing viscosity as rheopecty and thixotropy.

#Initialisms – The authors use the initialism RE before defining a couple paragraphs later.

We thank the reviewer for pointing out this oversight which we have now corrected.

#Linearizing Chains in Simulation – The main text very briefly describes the molecular model for digestion as "linearizing chains," but I think this is a bit too ambiguous. I think it would be appropriate for the authors to include a few more details in the main text that indicate what is meant by linearizing (ring opening + removal of torsional stiffness).

We agree that the main text would benefit from more details about the linearisation procedure. We have now added them and report an excerpt here:

"The linearisation is done by removing one bead from each cut ring together with its patches and all the bonds, angles and dihedrals that it is part of. Simultaneously, we zero all the torsional constraints along the cut chain, but leave the bending rigidity unaltered. This procedure is motivated by recent simulations showing that the twist relaxation of DNA is orders of magnitude faster than the relaxation of the writhe [...] This means that shortly after being cleaved by a restriction enzyme, DNA is likely torsionally relaxed, yet still displays a large amount of unresolved writhe, as in our simulations (Fig. 2)."

###Comments Regarding Modeling Methods In the Supplement

#Patch modeling -Please specify how “patches” are modelled in your bead-spring polymer model. Do your individual beads track an internal vector orientation, or are your patches modeled with additional beads affixed to the backbone beads with rigid-body constraints?

We thank the reviewer for requesting clarity on this point. We use the second approach. Patches are modelled as additional non-interacting beads that decorate the backbone. The backbone bead and the 3 patches behave as a rigid body with a damp parameter set to 4 such that its inertial time $t_{\text{inertia}} = \text{mass}/\text{damp} = 1$ is that of a bead of mass 1 (even though each body is made by 4 beads with mass 1). *We have now added these details in both the main text (Methods) and the SI.*

Patches and rotational inertia - Are the patches decorating the beads given mass to produce a moment of inertia along the backbone? If so, please specify. If not, do the authors observe any unusual behavior due to an absence of “torsional inertia.”

We thank the reviewer for the insightful question. The patches do have a mass. As mentioned above, we set the damp parameter such that the inertial time is 1 in reduced LJ units and so much shorter than the diffusive dynamics of the whole chain. We now note in the SI that the patches have mass.

#FENE Bond parameters -The choice of FENE bond potential parameters is just slightly different than the values typically used - $k = 30\epsilon/\sigma^2$ and $R_0 = 1.5\sigma$. Can the authors include a sentence in their methods rationalizing this slightly different choice? Is it to shift the average bond-length closer to 1σ ?

These were in fact typos, and we thank the referee for spotting them. The values of R_0 and k are indeed the usual 1.5σ and $30\epsilon/\sigma^2$.

Removing Torsion in Digested DNA – The authors use a torsional stiffness with a controlled pitch to drive ring DNA chains to twist. This torsion potential is removed when they want to study uncoiled DNA. I may be naïve to the mechanism of supercoiling, but I would appreciate the authors explaining why they completely zero-out the torsional stiffness when they model ring digestion. I understand the goal is to create a fraction of uncoiled DNA, but can the authors comment on their model versus the real system?

We thank the reviewer for raising this question. As mentioned above, recent computational evidence suggest that twist relaxation is orders of magnitude faster than writhe relaxation. This also makes sense for real DNA. Indeed, entanglements do not affect twist, so the twist relaxation is expected to be independent of the concentration of DNA. Because of this fact, and to simplify the computational implementation, we assume that after linearization the twist relaxes quickly, rendering the chain effectively twist free. Yet, writhe is much slower and its relaxation time is controlled by the degree of entanglements.

I'd imagine that the real system supercoils due to stored torsion in the ring backbone which has a torsional stiffness (twists along the chain) that cannot unwind due to the ring topology. Cutting the ring allows this stored torsion to unwind and shuts off the twist-bend instability. Is this a correct understanding?

Yes, the reviewer is correct. Although rather than “twist-bend instability”, the supercoils/plectonemic shapes are generated via the conservation of linking number (White-Fuller-Calugareanu WFC theorem).

We prepare the polymers as flat ribbons and then impose a twist. Thanks to the conservation of linking number between the edges of the ribbon, twist can interconvert into writhe while keeping the linking number constant. This is a phenomenon that has to do with the interplay of geometry and topology.

The “twist-bend instability” is a different phenomenon related to the coupling of bending constants and the asymmetry of the DNA double helix that states that when DNA bends it also twists, and viceversa. Instead, the WFC theorem holds for any closed ribbon/tube.

If so, then wouldn't the linear DNA still have a torsional stiffness, and could it have some influence on the conformations of the linear DNA? Have the authors examined the conformational statistics of the linear DNA with and without a torsional stiffness turned on?

We thank the reviewer for raising this insightful point. We have not checked if maintaining the torsional stiffness would change the statistics of the linear chains. We expect this effect to be minor, and negligible compared with the dominating role of entanglements, which are vastly different for linear and ring topologies. We are grateful for the suggestion though, and we will explore the role of torsion on linear chains in **dilute conditions** in future studies.

Aside from efficiency, is there any reason the authors couldn't model supercoiled DNA by pre-twisting their rings and allowing them to unwind?

No. The effect would be the same. It turns out that initialising them as flat ribbons makes it easier to create systems with different values of supercoiling starting from the same (flat) initial condition.

Reviewers' comments:

Reviewer #1 (Remarks to the Author):

The new version of the manuscript tries to address the major concerns raised by the Reviewer, both conceptual and technical ones. The authors have re-organized the paper, with an aim to address the concerns and, wherever relevant, have improved the clarity of their results and the corresponding experimental methods. Despite the efforts, unfortunately, some of the key concerns still persist, and others remain open and/or partially addressed. Given that all the concerns raised were key to the quality and final message of the paper, the Reviewer is not able to recommend publication of the revised manuscript. Below, is a gist of the key conceptual concerns which still persist:

- 1) How do the authors decouple the confounding factors which contribute to the rheology results? - here the authors have been able to correctly identify the confounded parameters, namely the degree of overlap, average size of the DNA coils, and topological signature; yet the revised presentations fail to report the specific link between topology and rheology as claimed by the authors. Both experiments and BD simulations offer geometric, rather than topological reasoning of the observed behaviours.
- 2) The second point related to the temporal resolution, generates inconsistencies with the claims of the author: on the one hand the authors employ a quasi steady-state analysis to interpret the MSD data; yet time-dependent MSD data as the topological nature of the fluid changes, is missing. In light of this, the authors' claims of a "topologically-active" and "programmable system" appear a bit stretched.
- 3) Insights from additional simulations: Additional simulation data suggest that the linearizing factor and the coil size are first order parameters which determine the rheological outcomes of the DNA-solutions. How topology - the central claim in this work - determines the rheological outcome is still not explicit.

Reviewer #2 (Remarks to the Author):

I appreciate the effort made by the authors to respond to my initial comments. New data included in Fig 1 and Fig 4 and the new structure of results have strengthened the paper. Unfortunately, I'm not convinced that the primary finding, that RE conversion of supercoiled DNA to linear topology leads to a monotonic increase in viscosity while DNA fragmentation decreases viscosity, represents a novel principle of controlling molecular topology and flow properties of DNA solutions. To me it appears the results are two pieces joined together that are either well-known (DNA fragmentation decreases viscosity) or can be directly derived from existing literature (the fact that chain contamination in ring polymers increases viscosity was reported in 1980s, e.g. Roovers *Macromolecules* 1988). For this reason, the programmable rheology seems intuitive in its current form. In view of the standard of the journal, I would look forward to a more counterintuitive principle or design of programmable rheology.

Reviewer #3 (Remarks to the Author):

I am satisfied with the changes and replies the authors have provided in response to this review. I recommend the Editor accept this manuscript for publication.

Reviewer #1 (Remarks to the Author):

The new version of the manuscript tries to address the major concerns raised by the Reviewer, both conceptual and technical ones. The authors have re-organized the paper, with an aim to address the concerns and, wherever relevant, have improved the clarity of their results and the corresponding experimental methods. Despite the efforts, unfortunately, some of the key concerns still persist, and others remain open and/or partially addressed. Given that all the concerns raised were key to the quality and final message of the paper, the Reviewer is not able to recommend publication of the revised manuscript. Below, is a gist of the key conceptual concerns which still persist:

1) How do the authors decouple the confounding factors which contribute to the rheology results? - here the authors have been able to correctly identify the confounded parameters, namely the degree of overlap, average size of the DNA coils, and topological signature; yet the revised presentations fail to report the specific link between topology and rheology as claimed by the authors. Both experiments and BD simulations offer geometric, rather than topological reasoning of the observed behaviours.

We thank the reviewer for recognizing that we now fully consider and discuss the different parameters that contribute to the rheology we measure. However, we are unsure how the reviewer missed our extensive discussion of decoupling these different factors in the revised manuscript. We are also unsure what the reviewer means by geometric rather than topological reasoning. By extensive investigation and dissection of the different 'confounding' parameters we show that the topological signature of the DNA indeed drives the time-varying rheology, and its effect depends in a programmable way on the initial DNA concentration (i.e. overlap).

In both simulations and experiments shown in Figs 1-3, the topology of the molecules is the *only* factor that is changing (via the action of incorporated enzymes) – going from supercoiled topology to linear topology. **It is quite striking that we measure such pronounced rheological changes in both experiments and simulations considering that the only action that is occurring is the in situ cutting of each DNA molecule at a single site to change its topology.** To determine the underlying mechanisms that give rise to the topology-driven time-varying rheology, we consider the effects of changing topology on the other properties of the molecules. Specifically, we investigate the effect of changing topology on polymer coil size. We explicitly delineate the effect of polymer overlap from the topological signature by performing experiments and simulations at varying DNA concentrations (and thus overlap). We show explicitly that the effect of changing topology (the rheopectic behavior) increases with increasing DNA overlap (i.e. concentration). DNA concentration and topology are thus two independent tuning knobs that can be used to program the non-equilibrium rheology.

We also explicitly consider and decouple the effect of changing topology on the DNA coil size in our BD simulations. We directly measure the radius of gyration and the mean end-to-end distance of the molecules at varying fractions of cut (linear) molecules. We show that both size metrics increase with increasing linearization of molecules. **This result is, in fact, nontrivial as there is currently no consensus on the effect of supercoiling on DNA coil size, nor is there an agreed upon method for estimating the size.** As supercoiling can lead to more compact structures as well as extended, rod like structures and even branched structures, the role of topology on DNA coil size is still a topic of fervent debate.

Finally, we point out that the results of Fig 5, which show rheopectic rheological behavior in the absence of multi-cutter enzymes, are using a different DNA size and concentration than those used for results shown in Figs 1-3. These results delineate the effects of DNA size and topology and show that our topologically-driven rheopectic behavior is robust to changing DNA size.

2) The second point related to the temporal resolution, generates inconsistencies with the claims of the author: on the one hand the authors employ a quasi steady-state analysis to interpret the MSD data; yet time-dependent MSD data as the topological nature of the fluid changes, is missing. In light of this, the authors' claims of a "topologically-active" and "programmable system" appear a bit stretched.

We strongly disagree with the reviewer that the claims are inconsistent or stretched and that the time-dependent MSD data is missing. **Most importantly, contrary to what the reviewer states, we do in fact show time-dependent MSD data in 3 of our 5 figures** (Figs 1c, 2b-c, 4c). We also show in the SI (Fig S1) six different plots of time-dependent MSDs (each showing MSDs for 8 different time points). It is deeply concerning that the referee missed all of these data and suggests that they did not give our revised manuscript a fair review.

In regards to our quasi steady-state analysis of MSDs, we specifically programmed the topological activity to be slow compared to the measurement timescale so that we could employ quasi steady-state analysis to interpret MSDs. We purposefully designed the system to have this separation of timescales to be able to unequivocally characterize the time-varying rheological properties based on established theories that relate MSDs to rheological properties. As our experiments and modeling results demonstrate, we (or others who adopt our platform) can design and program faster changes if we (or they) wish to by increasing the concentrations of the various restriction enzymes we employ. However, because there is no straightforward way to analyze non-steady-state MSDs to extract rheological properties, and because we wanted to ensure our analysis approach was robust and well established in the literature, we focused on systems that have this separation of timescales. While outside the scope of this work, we are currently working on developing theoretical descriptions of non-equilibrium MSDs and connecting them to rheological properties.

3) Insights from additional simulations: Additional simulation data suggest that the linearizing factor and the coil size are first order parameters which determine the rheological outcomes of the DNA-solutions. How topology - the central claim in this work - determines the rheological outcome is still not explicit.

We believe the reviewer has misinterpreted our new simulation results, which we unambiguously show and describe in the revised Fig 2. We are also puzzled by what the reviewer means by the 'linearizing factor'. **The data shown in Fig 2b-g explicitly and unequivocally show how DNA topology determines the rheological outcomes.** The parameter that we vary here (ϕ_L) is the fraction of DNA in the solution that is of linear topology versus supercoiled topology. This parameter is a direct and unambiguous metric of DNA topology: $\phi_L = 0$ is a solution in which all DNA molecules have supercoiled topology and $\phi_L = 1$ solutions comprise only linear topology DNA. The intermediate ϕ_L values have varying fractions of each topology. **The only parameter changing in these plots is the topology of the DNA—and there is an obvious impact on DNA diffusion and thus solution rheology—which we measure in experiments.**

As the reviewer notes, we also examine how changing topology impacts the size of the DNA coils, which we then use to quantitatively determine the scaling relation between the topology-dependent and time-dependent coil size and the resulting rheology. **No previous work has revealed these important connections, relations, and functional descriptions.**

Reviewer #2 (Remarks to the Author):

I appreciate the effort made by the authors to respond to my initial comments. New data included in Fig 1 and Fig 4 and the new structure of results have strengthened the paper. Unfortunately, I'm not convinced that the primary finding, that RE conversion of supercoiled DNA to linear topology leads to a monotonic increase in viscosity while DNA fragmentation decreases viscosity, represents a novel principle of controlling molecular topology and flow properties of DNA solutions. To me it appears the results are two pieces joined together that are either well-known (DNA fragmentation decreases viscosity) or can be directly derived from existing literature (the fact that chain contamination in ring polymers increases viscosity was reported in 1980s, e.g. Roovers *Macromolecules* 1988). For this reason, the programmable rheology seems intuitive in its current form. In view of the standard of the journal, I would look forward to a more counterintuitive principle or design of programmable rheology.

We are pleased that the reviewer appreciates our efforts and acknowledges that our new data and structure have indeed strengthened our paper. However, we respectfully yet strongly disagree with the reviewer that the results we present are either 'well-known' or 'can be directly derived'. As we detail below, it appears that the reviewer has deemed our results intuitive because they have not given our revised manuscript a fair read, resulting in a misunderstanding and/or a priori dismissal of our results.

Firstly, with the fragmentation of DNA, we acknowledge that one would expect to measure a decrease in viscosity as the DNA length is shortened. However, what is not at all intuitive is the functional form of this decrease and the parameters that can be used to program this functional dependence. **This problem has not been addressed in the literature and is critical to the design of programmable topologically-active materials.** To this end, we **developed new non-equilibrium living polymer theory** to universally describe the thixotropic behavior we measure, which we show is robustly applicable to all types and concentrations of multi-cutter enzymes employed. We also explicitly show, **for the first time**, the quantitative relationship between DNA fragment length (architecture) and rheology, and extract the universal digestion rate and its dependence on enzyme concentration. **These substantial theoretical and experimental advances are highly non-trivial and critical to building up even more complex and counterintuitive non-equilibrium rheology, such as multi-mode gated rheology we show in Fig 5.**

Regarding the results of our linearization experiments and simulations (Figs 1-3), we strongly disagree that the results we present could be derived from the existing literature. The papers that the reviewer cites here (and in their previous report) are all for *steady-state* blends of *ring* (i.e., relaxed circular) and linear polymers. Here, we are examining *non-equilibrium* conversion of *supercoiled* circular constructs to linear form. Supercoiled polymers are a much more topologically complex than their ring or linear counterparts, and there is a **paucity of data on the effect of supercoiling of polymers or DNA on rheology**. In fact, there is only one paper to our knowledge (from our own group) that considers the rheological properties of solutions of supercoiled DNA. In this prior work, the rheology of steady-state blends of supercoiled and ring DNA with varying overall concentrations was measured. There was no examination of how the relative fraction of each topology impacts the results, nor were blends of linear and supercoiled DNA examined. **Even more important**, in this previous work, the topology of the DNA was not changing over time, so the functional form of the time-dependent rheological properties – and how topological changes determine these non-equilibrium rheology – could not be determined or even inferred. Such knowledge and control, which we demonstrate here **for the first time** is critical to the design of topologically-controlled non-equilibrium materials.

If we instead consider the body of literature that the reviewer mentions on blends of rings (non-supercoiled relaxed circles) and linear polymers, then we would in fact **not** expect a monotonic decrease in viscosity (rheology), as threading of rings by linear chains has been shown to result in a non-monotonic dependence of viscosity on the fraction of linear chains in the blend, with a peak viscosity

reached when the relative concentrations of rings and linear chains are comparable. We describe (and properly cite) this phenomenon in the revised manuscript. Importantly, we measure no such peak, but instead measure a monotonic decrease in viscosity with time. This key difference between the topology-rheology relationship we show here and those previously reported, suggests that threading of supercoiled polymers by linear chains is much less pervasive than for ring polymers – **an important yet previously undiscovered result**. We also highlight that the **non-equilibrium nature of our system – representing a major advance in the field – reveals dynamic heterogeneities that arise from the continuously changing topological signature of the solutions** and their impact on the interactions between topologically distinct polymers. This effect had yet to be discovered and would be impossible to measure in steady-state experiments or predicted from models thereof—in which the polymer topology is not changing.

Finally, we point to the results of Fig 5 which demonstrate that our topologically-active approach to materials design can result in complex time-varying rheology with time-dependence and strength that can be precisely programmed by varying concentrations and types of enzymes – thereby introducing a new class of non-equilibrium materials that can be engineered to exhibit a wide range of rheological properties with programmable functional forms of the time-varying rheology directly determined by in situ enzymatic conversion of polymer topology.

Reviewer #3 (Remarks to the Author):

I am satisfied with the changes and replies the authors have provided in response to this review. I recommend the Editor accept this manuscript for publication.

We are pleased that the reviewer is satisfied with our changes and response and recommends publication in *Nature Communications*.

REVIEWER COMMENTS

Reviewer #4 (Remarks to the Author):

The main concept of this work, the demonstration of an activity-driven non-monotonic time evolution of the complex viscosity in entangled DNA fluids, is novel and potentially impactful. I may be published eventually in Nature Comm., however, in my opinion, some points ought to be carefully addressed:

1. Overall, the authors are urged to tone-down. Some of their many grand statements are unsubstantiated and, simply put, wrong. The key example is the rheopectic and thixotropic behavior which appear everywhere. Here we have an increase or a decrease of DYNAMIC (complex) viscosity with time but the material have not been interrogated by a (nonlinear) shear field. Hence, the non-monotonicity (the main result of the work) is convincing, but this is a result of changing structural/conformational behavior of the DNA over time with the induced enzymatic reactions, as the authors state. The only external field is the activity due to the enzymes. The authors are advised to consult the rheology literature (any textbook would do) and correct their manuscript accordingly.
2. The authors put too much emphasis on their materials, and understandably so. However, for such a work, proper characterization of the different structures requires more careful analysis by preparative electrophoresis. Since this is not done apparently, there is always a concern about the true structure of the experimental system. It is important to recognize the limitations.
3. Notwithstanding the above, the active manipulation of viscosity due to change in topology is an important result. There is an analogy to the topological mixtures of polymers (synthetic and DNA), where the viscosity of ring/linear polymer mixtures depends non-monotonically on the mass fraction. However, this has been observed by studying separate samples and at steady state. Here, one could achieve this with one sample by observing the time response via the enzymatic reactions. The authors have addressed this duality (similarities, differences) but it is important to emphasize more as this is the punchline of the work.

Reviewer #5 (Remarks to the Author):

This work reports on some elegant experiments and supporting simulations that show how the rheology of DNA solutions is affected by the action of topology-related enzymes. The rheological manifestations of topological properties of polymers attracted much attention in the literature. The interest did not diminish since De Gennes and Doi and Edwards worked out the reptation model in 1970ies, and the attention to the field grew even stronger in the last few years. It is now well established that the viscosity of a pure melt of unconcatenated rings is larger than that of pure linear chains, while very small contamination of rings with linear chains leads to drastic increase of viscosity. Present work very nicely supports and demonstrates this idea by taking DNA rings and slowly transforming them into linear ones by the action of restriction enzymes.

Since dsDNA supports not only bending, but also twisting deformation, there is an additional factor in DNA rheology-topology interplay, related to superhelices and plectonemes. I don't think the authors completely clarified this issue, but they did present some interesting data on it.

Although there are some relatively small technical issues with both experimental and

simulation parts of the work, they are not the main problem that I see with this work. The main problem that I see is that authors for some reason present their work as something conceptually entirely different than what it really is.

Authors start by declaring that topological alterations can drive systems out of equilibrium and that "topologically active systems" somehow represent a different class of active systems, distinct from the ones where activity is associated with dissipation of energy. I find this statement dangerously confusing. Of course, if a system is initially in a state of thermodynamic equilibrium subject to some sort of topological constraints, and then these constraints are relaxed, then the system can start its relaxation to another less restricted thermodynamic equilibrium. For instance, a dsDNA loop explores a set of accessible conformations, but once it is cut, it starts exploring a much wider set of conformations. This is quite trivial. However, to associate this with active systems is entirely misleading, for fundamental thermodynamic reasons. No topological manipulations without energy expenditure can drive the system anywhere except to relaxation.

This conceptual misstatement seems in line with other disproportional and exaggerated claims in the work. It is true that enzymatic cutting of DNA rings leads to change of viscosity -- but why is it "programmable"? It is definitely predictable and understandable, but it seems no more "programmable" than falling of a pencil on the floor after it is pushed over the edge of a desk. Similarly, when authors say that they want to "facilitate programming of rheological properties..." -- that simply means that they examine viscosity in its dependence on various parameters. It is certainly a useful task, but its significance is only diminished and confused by the pompous justification.

To summarize, I think that the authors did not do justice to their work trying to place it in the context of programmable active systems. I think that the work could be much stronger if it was presented for what it really is -- the study of rheology of DNA solutions under the action of restriction and other topology modifying enzymes. This itself is quite a large task, and authors could do a better job delineating aspects that are clear from what is still only tentative, such as the relative importance of various factors (coil size, chirality, density of helical superturns, etc) is determining viscosity.

To summarize, the work reports on potentially interesting experimental and computational study, but presentation of the results is marred with misplaced claims and even some misconceptions.

NCOMMS-21-44830B-Z Response to Reviewer Comments

Reviewer comments are reproduced in green font

Excerpts from the revised manuscript are in blue font. References are not included in the excerpts (replaced with [...]) but can be found in the manuscript.

Reviewer #4:

The main concept of this work, the demonstration of an activity-driven non-monotonic time evolution of the complex viscosity in entangled DNA fluids, is novel and potentially impactful. I may be published eventually in Nature Comm., however, in my opinion, some points ought to be carefully addressed:

We thank the reviewer for finding our work 'novel and potentially impactful', and suggesting suitability in Nature Communications after addressing their comments. We have carefully addressed all the points the reviewer raises below, which have served to strengthen our manuscript.

1. Overall, the authors are urged to tone-down. Some of their many grand statements are unsubstantiated and, simply put, wrong. The key example is the rheopectic and thixotropic behavior which appear everywhere. Here we have an increase or a decrease of DYNAMIC (complex) viscosity with time but the material have not been interrogated by a (nonlinear) shear field. Hence, the non-monotonicity (the main result of the work) is convincing, but this is a result of changing structural/conformational behavior of the DNA over time with the induced enzymatic reactions, as the authors state. The only external field is the activity due to the enzymes. The authors are advised to consult the rheology literature (any textbook would do) and correct their manuscript accordingly.

We thank the referee for this comment. We had used "thixotropy" and "rheopecty" to make contact with the relevant community of researchers interested in time-dependent rheology behaviours. However, we realize that the time-dependent change in viscosity we are measuring is not due to straining but to the internal action of the enzymes, so the terminology is not entirely accurate. As such, we have now adjusted the jargon throughout and substituted these terms with **time-dependent change in viscosity**, **time-varying viscosity**, **increase/decrease in viscosity** or similar phrasing. We have also changed the title to "**Topological digestion drives programmable time-varying rheology of entangled DNA fluids**". All instances of these changes are in blue font in the revised manuscript.

2. The authors put too much emphasis on their materials, and understandably so. However, for such a work, proper characterization of the different structures requires more careful analysis by preparative electrophoresis. Since this is not done apparently, there is always a concern about the true structure of the experimental system. It is important to recognize the limitations.

We appreciate the reviewer's concern regarding proper characterization of our materials. Because the constructs we use in this work have been extensively characterized and validated in previous works we did not include many details regarding their characterization in the present work. We included SI Fig S4 that detailed our quantitative characterization of the different DNA topologies and concentrations comprising our materials. Our extensive electrophoresis experiments, results, and analysis that we presented in the manuscript focused on characterization of the structures *during* enzymatic activity (Fig 1f-j, Fig 4j,k, SI Fig S3). To address the reviewer's comment, we have added the following clarifying statements to point to the careful characterization and analysis of our constructs and fluids:

'We use pYES2 as our substrate as its characterization and purification have been thoroughly described and validated in previously published works [...] (also see SI Fig. S4), such that we can be confident in the initial state of the fluids. Likewise, BamHI is an extensively-used inexpensive RE with validated single-cutting action on pYES2 [...].'

'pYES2 and the purification and characterization methods we use have been thoroughly described and validated in the literature [...].'

'Importantly, this commercially available DNA construct has been used for decades as a model polymer system and has been exhaustively characterized in the literature [...], such that we can be confident of the initial properties of the fluid. Moreover, our analytical gel electrophoresis experiments (Fig 4j) show that both the initial DNA length, as well as the number and lengths of the digested fragments are as expected, further indicating the structural purity of our fluids.'

'We note that this construct has been used for decades as a model polymer system and its characterization has been published extensively [...].'

We do not use preparative electrophoresis to purify our constructs as the concentration and quantity of DNA possible to extract using this method is prohibitively small. We do, however, use analytical gel electrophoresis to quantitatively determine the concentrations of the different topologies and use molecular markers to confirm the topology of the various bands. Nevertheless, as suggested by the reviewer, we have removed some of the emphasis we place on materials applications.

3. Notwithstanding the above, the active manipulation of viscosity due to change in topology is an important result. There is an analogy to the topological mixtures of polymers (synthetic and DNA), where the viscosity of ring/linear polymer mixtures depends non-monotonically on the mass fraction. However, this has been observed by studying separate samples and at steady state. Here, one could achieve this with one sample by observing the time response via the enzymatic reactions. The authors have addressed this duality (similarities, differences) but it is important to emphasize more as this is the punchline of the work.

We are very happy to see that the reviewer appreciates one of our main points. Indeed, we were pleased to discover that we could tune the ring/linear composition 'in one pot' via enzymatic reactions. We have added to the text to emphasize this important point, as follows:

'Our approach of harnessing topological conversion to drive time-dependent rheological behaviours builds on recent efforts exploring the connection between polymer topology and fluid rheology [...]. Notably, our results shed important new light on the impact of supercoiling of ring polymers on the rheology of ring polymer solutions and ring-linear blends. Moreover, our strategy allows for efficient and precise determination of the dependence of rheology on the relative concentrations of topologically-distinct polymers comprising blends. For example, in typical experiments that examine how the ratio of rings and linear chains impacts the rheology of ring-linear blends, each blend is 'man-made' by mixing the two solutions comprising the distinct topologies at a specific ratio, such that each data point is from a different sample in a different chamber. This process is not only susceptible to error through concentration variations and mixing inconsistencies, but the number of different topological ratios that can be investigated is limited by available sample and extended data acquisition times.

With our approach, a continuum of topological ratios is achieved via in situ RE digestion of a single sample. By tuning the timescale of digestion to be slow (several hours) compared to the measurement time (minutes), we are able to achieve remarkably high resolution in formulation space, measuring the viscosity at, e.g., 24 different blend ratios over the course of four hours. Extending the measurement

window and/or slowing the digestion rate further can allow for even greater precision.'

'Finally, our strategy allows for efficient and precise determination of the dependence of rheology on the relative concentrations of topologically-distinct polymers comprising blends - achieving remarkably high resolution in formulation space via in situ RE digestion of a single sample.'

Reviewer #5 (Remarks to the Author):

This work reports on some elegant experiments and supporting simulations that show how the rheology of DNA solutions is affected by the action of topology-related enzymes. The rheological manifestations of topological properties of polymers attracted much attention in the literature. The interest did not diminish since De Gennes and Doi and Edwards worked out the reptation model in 1970ies, and the attention to the field grew even stronger in the last few years. It is now well established that the viscosity of a pure melt of unconcatenated rings is larger than that of pure linear chains, while very small contamination of rings with linear chains leads to drastic increase of viscosity. Present work very nicely supports and demonstrates this idea by taking DNA rings and slowly transforming them into linear ones by the action of restriction enzymes.

We thank the reviewer for finding our work 'elegant' and noting that it 'very nicely supports and demonstrates' important concepts in polymer physics.

Since dsDNA supports not only bending, but also twisting deformation, there is an additional factor in DNA rheology-topology interplay, related to superhelices and plectonemes. I don't think the authors completely clarified this issue, but they did present some interesting data on it.

We thank the reviewer for finding our data on supercoiled DNA interesting and distinct from open circular (ring) polymers. As well argued by the reviewer, supercoiling is an aspect that cannot be studied by experiments with synthetic polymers that have no torsional rigidity, adding to the uniqueness of our system. Indeed, our results suggest that it is the supercoiling of the circular DNA that leads to the monotonic increase in viscosity during enzymatic linearization, owing to the fact that supercoiling leads to more compact conformations compared to open rings that substantially swell when linearised. Additionally, previous evidence suggest that supercoiled DNA plasmids are not as easily threaded as open ring polymers. Further support of this fact is our data shown in Fig 1 that indicates increased prevalence of threading at later times in the digestion when the fluid is comprised primarily of open rings and linear chains. We make these points clear in the revised manuscript as follows:

'Importantly, the increase in viscosity shown in Fig. 1k,l is generally monotonic over the course of the digestion time, counter to the non-monotonic dependence of viscosity reported in ring-linear polymer blends [...]. However, the trend is 'noisy' with dips and valleys for all but the lowest DNA concentration, as well as larger standard deviations in the data at each time point. Increased dynamical heterogeneity has been previously predicted and observed in entangled rings and ring-linear blends [...] and attributed, in part, to ring-ring and ring-linear threading events [...]. Other features of our data that support the contribution of threading to the rheology are the steep upticks in the viscosity for the (6 mg/ml, 0.05 U/ μ g), (3 mg/ml, 0.05 U/ μ g) and (3 mg/ml, 0.10 U/ μ g) cases which occur at times that correspond to ~15% and ~70% linear chains. The low linear fraction is comparable to the fraction of linear chains needed in ring polymers to markedly reduce ring diffusion via threading, while the high fraction is similar to that previously reported as necessary for a large increase in complex viscosity [...].'

'Further, our gel electrophoresis analysis (Fig. 1f-h, SI Fig. S3) shows that the supercoiled DNA is digested

at a faster rate than the ring DNA, such that we expect ring-linear threading to contribute more to our observations at later times in the digestion. Indeed, the features described above are most apparent at these later times. Finally, we note that while threading of open rings has been well-established, the literature regarding the extent to which supercoiled polymers can become threaded is scarce and lacks consensus [...].'

'Simulations do show a slightly weaker dependence of D/D_0 on ϕ_L and concentration, as compared to our experimental data, which we conjecture arises from the lack of open rings in the simulations. Recall that our experimental solutions start out with both supercoiled and open rings, with the rings appearing to be cut at a slower rate (Fig. 1f-h). As such, we attribute the less pronounced slowing down in the simulations as compared to experiments to fewer threading events, which have been shown to markedly slow dynamics [...].'

Although there are some relatively small technical issues with both experimental and simulation parts of the work, they are not the main problem that I see with this work. The main problem that I see is that authors for some reason present their work as something conceptually entirely different than what it really is.

Authors start by declaring that topological alterations can drive systems out of equilibrium and that "topologically active systems" somehow represent a different class of active systems, distinct from the ones where activity is associated with dissipation of energy. I find this statement dangerously confusing. Of course, if a system is initially in a state of thermodynamic equilibrium subject to some sort of topological constraints, and then these constraints are relaxed, then the system can start its relaxation to another less restricted thermodynamic equilibrium. For instance, a dsDNA loop explores a set of accessible conformations, but once it is cut, it starts exploring a much wider set of conformations. This is quite trivial. However, to associate this with active systems is entirely misleading, for fundamental thermodynamic reasons. No topological manipulations without energy expenditure can drive the system anywhere except to relaxation.

We thank the reviewer for the feedback regarding how to better frame our work. We agree with the reviewer that our system does not represent an active system in the traditional sense. As the reviewer aptly describes, our systems are 'out-of-equilibrium' in the sense that they are 'en-route' from one equilibrium state to a new one. To avoid confusion on this point we have removed the first paragraph of the Introduction that framed our work in the context of active matter. We have also removed other instances that focus on connections between our work and other active matter systems.

However, it is important to point out that it is exactly this non-equilibrium relaxation that we aim to exploit to drive time-dependent rheology of the DNA fluids. The concentrations and types of the enzymes and DNA, as well as the size and topology of the DNA in the initial state can be precisely tuned to elicit a broad range of rheological alterations during the relaxation, and reach new states with rheological properties that are also programmable. We have now made these points clearer and more precise throughout the text to avoid confusion. Some excerpts include:

'Finally, we emphasize that the fluids we explore in this work are pushed out-of-equilibrium by an irreversible change in their topology, such that they are en-route from an initial equilibrium state to a new one. It is, in fact, this thermodynamic relaxation that we exploit to drive time-dependent alterations to the rheological properties of the DNA fluids. While the final state is known, and, like the initial state, is in thermodynamic equilibrium, the concentrations of the enzymes and DNA, as well as the size and topology of the DNA in the initial state can be precisely tuned to elicit a broad range of alterations to the rheological properties of the fluids that can occur on tunable timescales from minutes to hours. In this

way, we can program the time over which the system remains out-of-equilibrium, en-route to a new equilibrium state.'

'Here, we have studied a class of polymeric fluids that are pushed out-of-equilibrium by dissipative topological alterations to their architecture. More specifically, we have realised DNA fluids in which restriction enzymes cut DNA to trigger the relaxation of the system from one equilibrium state to another. As such, our fluids are en-route between two equilibrium states, during which they exhibit a wide range of time-varying rheological properties which depend on the parameters of our systems in a tunable manner. [...].'

This conceptual misstatement seems in line with other disproportional and exaggerated claims in the work. It is true that enzymatic cutting of DNA rings leads to change of viscosity -- but why is it "programmable"? It is definitely predictable and understandable, but it seems no more "programmable" than falling of a pencil on the floor after it is pushed over the edge of a desk.

While we appreciate the reviewer's point of view regarding programmability, we respectfully disagree. While the action of each particular enzyme on a specific DNA construct is indeed fixed and not programmable, the type of topological conversion, the number and type of cuts, and the lengths of the digested DNA fragments, can be precisely programmed by the choice of DNA construct and enzyme (of which there is an abundance of commercially available candidates).

More importantly, when we refer to our systems as "programmable", we are not referring to the enzymatic cutting process itself, but rather the impact of an ensemble of enzymes on the bulk rheology of dense DNA systems, which is, indeed, programmable even for a single enzyme type.

By varying the DNA and enzyme concentrations, enzyme type, and DNA size and initial topology, we can precisely tune the degree to which the viscosity of the fluid changes, and the timescales over which the rheological changes occur. All this, while keeping the underlying physics unchanged.

Moreover, by using a combination of enzymes we can engineer non-monotonic viscosity changes in which we can program the time at which peak viscosity is reached and the degree of increase/decrease by varying the concentrations of the two enzymes (Fig 5).

Finally, it is important to point out that the connection between changing topology and changing rheology is not predictable nor trivial. In fact, this is a key point that our data in Fig 1 shows: at a fixed enzyme:DNA stoichiometry in which the rate of topological conversion is fixed (as shown in Fig 1j), the effect on the rheology depends strongly on the concentration of DNA (Fig 1e,l). Specifically, unless the molecules are substantially overlapping, topological conversion has minimal impact on the rheology, and increasing the degree of polymer overlap leads to more dramatic increase in viscosity. This is a key demonstration of the programmability of our system that is independent of tuning the type or rate of enzymatic digestion.

We have revised the text in various locations to make these points clearer, as follows:

'The rich and surprising behavior of topologically-distinct polymeric fluids, along with the principal role that topology plays in dictating the conformational size, structure, and interactions of the comprising polymers, inspired us to exploit DNA topology and DNA-cutting enzymes as a route to functionalize polymeric fluids with rheological properties that vary over time with programmable rates.'

'In the preceding sections, we demonstrate that the action of REs can either increase and decrease the viscosity of entangled DNA fluids over time, depending on the initial DNA topology and type of RE. We found that the rate and degree to which the viscosity changes over time can be programmed by varying

the DNA concentration and topology as well as the RE concentration and type.'

'We now leverage these findings to design systems that can exhibit non-monotonic time-varying rheology. For instance, we aim to program DNA fluids to display an initial increase of viscosity, followed by a decrease of viscosity over time. Additionally, we aim to demonstrate that we can program the rate and magnitude of viscosity increase/decrease, as well as the time of the peak viscosity, referred to as 'gating time.'

'Finally, we emphasize that the fluids we explore in this work are pushed out-of-equilibrium by an irreversible change in their topology, such that they are en-route from an initial equilibrium state to a new one. It is, in fact, this thermodynamic relaxation that we exploit to drive time-dependent alterations to the rheological properties of the DNA fluids. While the final state is known, and, like the initial state, is in thermodynamic equilibrium; the concentrations of the enzymes and DNA, as well as the size and topology of the DNA in the initial state can be precisely tuned to elicit a broad range of alterations to the rheological properties of the fluids that can occur on tunable timescales from minutes to hours. In this way, we can program the time over which the system remains out-of-equilibrium, en-route to a new equilibrium state.'

Similarly, when authors say that they want to "facilitate programming of rheological properties..." -- that simply means that they examine viscosity in its dependence on various parameters. It is certainly a useful task, but its significance is only diminished and confused by the pompous justification.

We thank the reviewer for recommending rephrasing certain statements in the manuscript, which we have now done. As to the specific phrase the reviewer mentions, what we meant was that we aimed to explore and map the phase space of possible rheological alterations possible by varying the enzyme type and concentration as well as the concentration, length and topology of the DNA. This mapping, may, in the future, aid researchers in designing fluids with specific rheological properties for, e.g., bioprinting. From our perspective, this is what "facilitate programming/design" is. Nevertheless, we have clarified the meaning of this statement and similar ones to avoid diminishing the significance of our work. For example:

'The topologically-driven time-dependent rheological properties we demonstrate here may be exploited for applications in time-controlled targeted drug release and autonomous tissue healing at a wound site. However, future applications of DNA-based materials will require a better understanding of the relationships between the architecture of the DNA constituents and their bulk rheological properties [...], which our work aims to contribute to.

To summarize, I think that the authors did not do justice to their work trying to place it in the context of programmable active systems. I think that the work could be much stronger if it was presented for what it really is -- the study of rheology of DNA solutions under the action of restriction and other topology modifying enzymes. This itself is quite a large task, and authors could do a better job delineating aspects that are clear from what is still only tentative, such as the relative importance of various factors (coil size, chirality, density of helical superturns, etc) is determining viscosity.

We thank the reviewer for the suggestions. We have revised the manuscript to make clearer the key findings and their significance and give better justice to our work. Indeed, much of the introduction now gives a better overview of the state-of-the-art and open questions in the physics of polymers with non-trivial topology, rather than non-equilibrium, programmable fluids. Additionally, in the conclusions, we have added a paragraph to comment more precisely on the aspects that our work does not capture in full. Some examples of the amended text include:

'Our approach of harnessing topological conversion to drive time-dependent rheological behaviours builds on recent efforts exploring the connection between polymer topology and fluid rheology [...].'

'Notably, our results shed important new light on the impact of supercoiling of ring polymers on the rheology of ring polymer solutions and ring-linear blends. Moreover, our strategy allows for efficient and precise determination of the dependence of rheology on the relative concentrations of topologically-distinct polymers comprising blends. For example, in typical experiments that examine how the ratio of rings and linear chains impacts the rheology of ring-linear blends, each blend is 'man-made' by mixing the two solutions comprising the distinct topologies at a specific ratio, such that each data point is from a different sample in a different chamber [...]. This process is not only susceptible to error through concentration variations and mixing inconsistencies, but the number of different topological ratios that can be investigated is limited by available sample and extended data acquisition times. With our approach, a continuum of topological ratios is achieved via in situ RE digestion of a single sample. By tuning the timescale of digestion to be slow (several hours) compared to the measurement time (minutes), we are able to achieve remarkably high resolution in formulation space, measuring the viscosity at, e.g., 24 different blend ratios over the course of four hours. Extending the measurement window and/or slowing the digestion rate further can allow for even greater precision.'

'Finally, our strategy allows for efficient and precise determination of the dependence of rheology on the relative concentrations of topologically-distinct polymers comprising blends - achieving remarkably high resolution in formulation space via in situ RE digestion of a single sample.'

To summarize, the work reports on potentially interesting experimental and computational study, but presentation of the results is marred with misplaced claims and even some misconceptions.

We are pleased that the reviewer finds our work interesting, and appreciate their suggestions regarding the presentation of our work. As detailed above we have thoroughly addressed the reviewer's comments in the revised manuscript.

REVIEWER COMMENTS

Reviewer #4 (Remarks to the Author):

The manuscript presents a nice set of experiments, supported by simulations, that nicely complements the known concept of non-monotonic composition-dependent ring-linear viscosity due to threading. The elements of novelty with the observation and potential control of the phenomenon in one pot are evident. So, this is an advance in the field and in the particular are of DNA-based materials science. There have been three main problems:

a) Mistakes in the text, primarily due to the fact that the authors are not rheology experts. This has been fixed.

b) Issues with DNA characterization: whereas this has been addressed, often referring to literature, it remains a concern, but to be fair, the same holds for other experimental systems.

c) The tone of the presentation. This over-selling, typical of this group, creates confusion and reflects an unnecessary urge to promote their work, often leading to opposite results and confusion. Hopefully, the authors understand that this attitude does not serve them well. I see that they have tried to put their work better into context.

I conclude that the key issues of concern are addressed at satisfactory level, in my opinion. Hence, I recommend publication.

Reviewer #5 (Remarks to the Author):

Authors made significant efforts to address the concerns raised in the previous reports. I must admit, unfortunately, that I did not find the revisions entirely satisfactory. I will give here three examples:

(1) In the very first paragraph of the revised manuscript, authors say: "DNA is ... is kept out-of-equilibrium by proteins that continuously change its structure ..." and then a few lines later "... restriction endonucleases are able ... without consuming chemical energy". I maintain that this squarely contradicts thermodynamics. Nothing can continuously keep DNA (or anything else) out of equilibrium without energy expenditure.

(2) Closer to the end of the conclusions, authors say (in the newly written part of the text) "The topologically-driven time-dependent rheological properties we demonstrate here may be exploited for applications in time-controlled targeted drug release and autonomous tissue healing at a wound site." The distance from the study of time dependent viscosity of DNA solutions in the presence of enzymes to the treatment of wounds is enormous -- and this is still an understatement. To me this kind of unbounded speculation dramatically reduces the appeal of the paper rather than enhancing it.

(3) I found that the author's insistence on their use of "programmability" is just another example of unwarranted speculation. We call a computer programmable, but a set of vegetables in a supermarket is hardly deserving of this name, even though one can manipulate it to cook a variety of very different soups.

NCOMMS-21-44830C Response to Reviewer Comments

Reviewer Comments are reproduced in green font

Excerpts from the revised manuscript are in blue font. References are not included in the excerpts (replaced with [...]) but can be found in the manuscript.

REVIEWER COMMENTS

Reviewer #4 (Remarks to the Author):

The manuscript presents a nice set of experiments, supported by simulations, that nicely complements the known concept of non-monotonic composition-dependent ring-linear viscosity due to threading. The elements of novelty with the observation and potential control of the phenomenon in one pot are evident. So, this is an advance in the field and in the particular area of DNA-based materials science. There have been three main problems:

- a) Mistakes in the text, primarily due to the fact that the authors are not rheology experts. This has been fixed.
- b) Issues with DNA characterization: whereas this has been addressed, often referring to literature, it remains a concern, but to be fair, the same holds for other experimental systems.
- c) The tone of the presentation. This over-selling, typical of this group, creates confusion and reflects an unnecessary urge to promote their work, often leading to opposite results and confusion. Hopefully, the authors understand that this attitude does not serve them well. I see that they have tried to put their work better into context.

I conclude that the key issues of concern are addressed at satisfactory level, in my opinion. Hence, I recommend publication.

We are pleased that the Reviewer recommends publication.

Reviewer #5 (Remarks to the Author):

Authors made significant efforts to address the concerns raised in the previous reports. I must admit, unfortunately, that I did not find the revisions entirely satisfactory. I will give here three examples.

We are pleased that the Reviewer recognizes the 'significant efforts' we made to 'address their concerns.' In the revised version we have addressed the three remaining comments described below.

(1) In the very first paragraph of the revised manuscript, authors say: "DNA is ... is kept out-of-equilibrium by proteins that continuously change its structure ..." and then a few lines later "... restriction endonucleases are able ... without consuming chemical energy". I maintain that this squarely contradicts thermodynamics. Nothing can continuously keep DNA (or anything else) out of equilibrium without energy expenditure.

To address the Reviewer's comment, we have rewritten the first paragraph of the Introduction that includes the referenced statements as follows:

DNA is a model polymeric material that naturally occurs in multiple topologies, including supercoiled circular, relaxed circular (ring) and linear conformations. In cells, enzymes convert DNA from one topology to another to perform diverse biological functions...Specifically, type II restriction

endonucleases cleave the DNA backbone at specific restriction sites [...] by acting as catalysts to break the sugar-phosphate DNA backbone at specific sites.

(2) Closer to the end of the conclusions, authors say (in the newly written part of the text) "The topologically-driven time-dependent rheological properties we demonstrate here may be exploited for applications in time-controlled targeted drug release and autonomous tissue healing at a wound site." The distance from the study of time dependent viscosity of DNA solutions in the presence of enzymes to the treatment of wounds is enormous -- and this is still an understatement. To me this kind of unbounded speculation dramatically reduces the appeal of the paper rather than enhancing it.

To address the Reviewer's comment, we have removed this statement and rewritten the section as follows:

The topologically-driven time-dependent rheological properties we demonstrate here, along with the relationships that we elucidate between the architecture of the DNA constituents and their bulk rheological properties, may facilitate future materials applications.

(3) I found that the author's insistence on their use of "programmability" is just another example of unwarranted speculation. We call a computer programmable, but a set of vegetables in a supermarket is hardly deserving of this name, even though one can manipulate it to cook a variety of very different soups.

As suggested by the Reviewer, we have removed all instances of 'programmability', 'program' and 'programmable' in the revised manuscript.